# HyenaDNA: Long-Range Genomic Sequence Modeling at Single Nucleotide Resolution

**Eric Nguyen**[*,1]**, Michael Poli**[*,1]**, Marjan Faizi**[2,*]**,**
Armin W. Thomas[1], Callum Birch Sykes[3], Michael Wornow[1], Aman Patel[1],
Clayton Rabideau[3], Stefano Massaroli[4], Yoshua Bengio[4], Stefano Ermon[1],
Stephen A. Baccus[1,†], Christopher Ré[1,†]

## Abstract

Genomic (DNA) sequences encode an enormous amount of information for gene regulation, protein synthesis, and numerous other cellular properties. Similar to natural language models, researchers have proposed foundation models in genomics to learn generalizable features from unlabeled genome data that can then be fine-tuned for downstream tasks such as identifying regulatory elements. Due to the quadratic scaling of attention, previous Transformer-based genomic models have used 512 to 4k tokens as context (<0.001% of the human genome), significantly limiting the modeling of long-range interactions in DNA. In addition, these methods rely on tokenizers or fixed k-mers to aggregate meaningful DNA units, losing single nucleotide resolution (i.e. DNA "characters") where subtle genetic variations can completely alter protein function via single nucleotide polymorphisms (SNPs). Recently, Hyena, a large language model based on implicit convolutions was shown to match attention in quality while allowing longer context lengths and lower time complexity. Leveraging Hyena's new long-range capabilities, we present HyenaDNA, **a genomic foundation model** pretrained on the human reference genome with **context lengths of up to 1 million tokens at the single nucleotide-level** – an **up to 500x increase** over previous dense attention-based models. HyenaDNA scales sub-quadratically in sequence length (training up to 160x faster than Transformer), **uses single nucleotide tokens**, and has **full global context at each layer**. We explore what longer context enables - including the first use of in-context learning in genomics for simple adaptation to novel tasks without updating pretrained model weights. On a long-range species classification task, HyenaDNA is able to effectively solve the challenge by increasing the context length to 1M without downsampling. On fine-tuned benchmarks from the Nucleotide Transformer, HyenaDNA reaches state-of-the-art (SotA) on 12 of 18 datasets using a model with orders of magnitude less parameters and pretraining data.[2] On the GenomicBenchmarks, HyenaDNA surpasses SotA on 7 of 8 datasets on average by +10 accuracy points, and by as much as +20 accuracy points on enhancer identification. Code available at https://github.com/HazyResearch/hyena-dna.

---

[*]Equal contribution. † Equal senior authorship. [1]Stanford University. [2]Harvard University. [3]SynTensor. [4]Mila and Université de Montréal.

[2]On benchmarks from Nucleotide Transformer, HyenaDNA uses a model with 1500x fewer parameters (2.5B vs 1.6M) and 3200x less pretraining data (3202 vs 1 human reference genome).

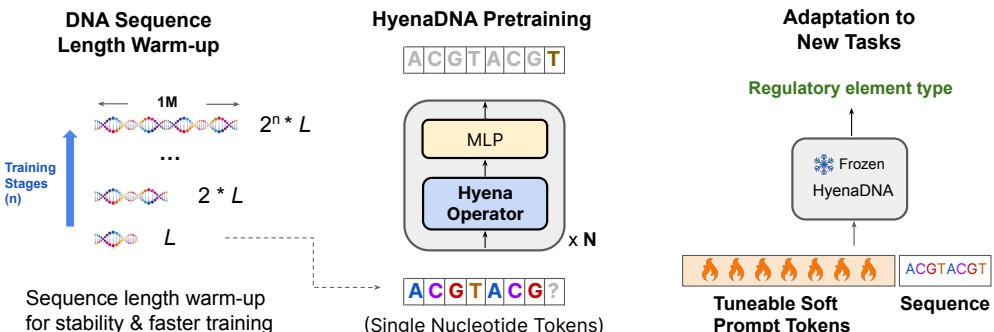

Figure 1.1: HyenaDNA recipe for long-range foundation models in genomics. The HyenaDNA architecture is a simple stack of Hyena operators [37] trained using next token prediction. (See Fig. 1.3 for block diagram of architecture). We introduce a new sequence length scheduling technique to stabilize training, and provide a method to leverage the longer context length to adapt to novel tasks without standard fine-tuning by filling the context window with learnable soft prompt tokens.

# 1 Introduction

Understanding and learning from DNA sequences has long been a goal of biologists and deep learning researchers, as its "language" encodes instructions essential for all living things [16]. The mapping from DNA instructions, genotypes, to observable function and traits, phenotypes, remains ongoing research effort. Towards this goal, researchers have proposed using foundation models (FMs) in genomics to learn generalizable features from unstructured whole genome data that can then be fine-tuned for a number of tasks including predicting the location and function of genes, identifying regulatory elements, and analyzing the evolution of species [25, 10, 20, 3, 53, 58]. In contrast to protein sequences, which have had successes in protein language models [29, 31, 32, 17, 5, 41, 14], DNA sequences are orders of magnitudes longer (e.g. the human genome is 3.2B nucleotides) with long-range dependencies and interactions that span over 100k+ nucleotides in length [1]. Overcoming the long-range limitations of current generation models could help drive the next wave of innovations in AI-powered drug discovery and therapeutics, and enable genomic FMs to understand and learn in-context whole patient genomes in a personalized way.

**Limitations of current models** Previous genomic FM approaches have relied on attention-based Transformers [25, 10, 53, 58], but face a number of challenges unique to DNA sequences. The attention mechanism scales quadratically in sequence length, with current genomic FMs pretraining on only 512 to 4,096 tokens as context [25, 58, 10, 55], <0.001% of the human genome. Also prevalent is the reliance on fixed k-mers, akin to DNA "words", and tokenizers to aggregate meaningful DNA units. However, single nucleotide alterations represent physical analogs where, for example, single nucleotide polymorphisms (SNPs) and mutations can have a profound impact on biological properties including regulatory activity [33]. In contrast, natural language semantics can often be conserved when single character or word changes occur over very long contexts. Therefore, having both **long-range context** and **single nucleotide resolution** simultaneously is critical, and remains a particular challenge in genomics.

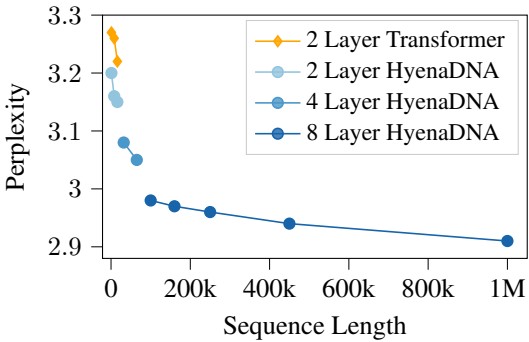

Figure 1.2: Pretraining on the human reference genome using longer sequences leads to better perplexity (improved prediction of next token).

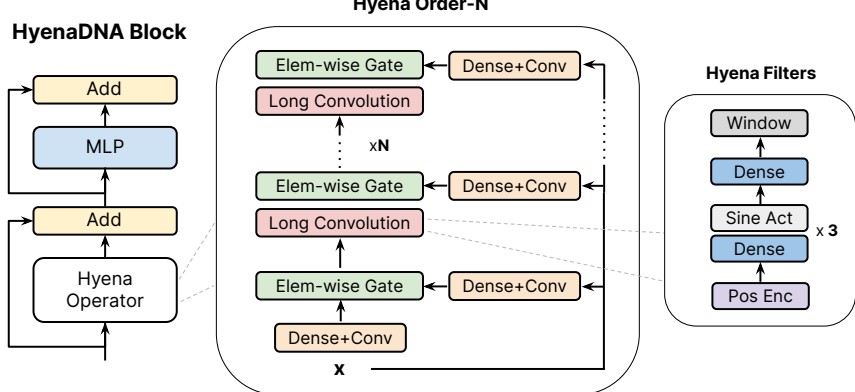

Figure 1.3: HyenaDNA block architecture. A Hyena operator is composed of long convolutions and element-wise gate layers. The gates are fed projections of the input using dense layers and short convolutions. The long convolutions are parameterized *implicitly* via an MLP that produces the convolutional filters. The convolution itself is evaluated using a Fast Fourier Transform convolution with time complexity $\mathcal{O}(L \log_2 L)$.

**Toward longer context models**   Recently, Hyena [37], a large language model based on implicit convolutions, was shown to match attention in quality while reducing computational time complexity, thereby allowing a longer context to be processed. Hyena uses a parameter-efficient **global convolutional filter** along with a **data-controlled gating** mechanism, which enables a context-specific operation over every token. Indeed, Hyena showed that for simple associative recall tasks using *synthetic* data, a shallow 2 layer model could effectively process context lengths at 131k tokens. We hypothesize that Hyena's core operations can unlock the potential to capture both the long-range and single nucleotide resolution of *real* genomic sequences over attention-based approaches. To test this, we explore two questions: **(i.) Can a convolutional long-context model be used effectively at single nucleotide resolution? (ii.) What new capabilities could long-context genomic foundations models enable?**

**HyenaDNA**   The result of our investigation is HyenaDNA, a genomic FM pretrained on the human reference genome at **context lengths up to 1 million tokens at single nucleotide resolution** - an up to **500x increase** over existing genomic FMs using dense-attention. HyenaDNA scales subquadratically in sequence length (training up to 160x faster than attention at sequence length 1M), uses single nucleotide tokens, and has a global receptive field at each layer. Our contributions include a "full-stack" recipe for building genomic FMs, including architecture design, a warm-up schedule to speed up training on ultralong sequences, and an efficient downstream adaptation procedure based on soft prompting and in-context learning.

**Full-stack genomics modeling**   We start with a decoder-only Hyena architecture pretrained using next nucleotide (token) prediction. We forego standard aggregating tokenizers, using a single-character tokenizer and a minimal DNA vocabulary of 4 nucleotides (plus special tokens). Training stability becomes an issue at ultralong sequences (200k+). To overcome this issue, we introduce a sequence length warm-up scheduler that gradually increases sequence length in stages. At sequence length 450k, training time is reduced by 40%, while boosting accuracy by 7.5 accuracy points on a species classification task. Furthermore, we design downstream adaptation procedures to leverage longer context windows, as simpler and more flexible alternatives to standard fine-tuning in genomics. This includes a novel soft prompt technique where learnable tokens (up to 32k) are injected directly into the input sequence itself, enabling competitive downstream results without the need to update a pretrained model.

**Genomic downstream tasks**   We apply our pretrained HyenaDNA models to 29 diverse downstream genomic tasks to showcase its long-range ability as well as fine-grain resolution. On fine-tuned benchmarks from the Nucleotide Transformer [10], HyenaDNA achieves state-of-the-

art (SotA) on 12 of 18 datasets while using a model with orders of magnitude less parameters and pretraining data (see Tab. 4.2). On the GenomicBenchmarks [23], HyenaDNA surpasses SotA on 7 of 8 datasets on average by +10 accuracy points, and by as much as +20 accuracy points on enhancer function identification. On a novel species classification task, HyenaDNA effectively solves the challenge by increasing the context length to 1 million tokens. In a challenging chromatin profile experiment, a 919-way multi-task, HyenaDNA performs competitively against a larger SotA sparse-attention BigBird Transformer [55]. Finally, we analyze the learned embeddings of a pretrained HyenaDNA model by clustering sequences by biotype (gene or transcription type) and compare the results with existing genomic FMs, showing that HyenaDNA can serve as an effective universal featurizer in genomics.

## 2    Preliminaries and Related Work

### 2.1    Transformers and Attention

Powering many recent *foundation models* is the *attention* mechanism. Given a length-$L$ sequence $x \in \mathbb{R}^{L \times D}$, a (single-headed) layer of *scaled self-attention* [2, 51] is a map from $\mathbb{R}^{L \times D}$ to $\mathbb{R}^{L \times D}$ which performs the following operations:

$$\mathsf{A}(x) = \sigma(x\mathsf{W}_q\mathsf{W}_k^\top x^\top), \quad y = \mathsf{A}(x)x\mathsf{W}_v \tag{2.1}$$

where $D$ is the embedding dimension, $\mathsf{W}_q, \mathsf{W}_k, \mathsf{W}_v \in \mathbb{R}^{D \times D}$ are learnable linear maps and $\sigma$ indicated row-wise softmax (and optional scaling). Attention computes all pair-wise comparison for every token, and scales as $\mathcal{O}(L^2)$ in sequence length. This allows a global context at high resolution, but limits the size of the context on current hardware.

Previous methods to reduce the quadratic cost of attention have used specialized methods to approximate full dense attention [18]. In sparse attention, elements attend only to a subset of all other positions. Alternatively, linear attention methods construct approximations to $\mathsf{A}(u)$ that can be evaluated in subquadratic time. Both of these classes of methods, however, trade lower time complexity (allowing longer sequences) for loss in expressivity.

### 2.2    Long Context Strategies in Genomics

To achieve longer context, genomic models have relied on two strategies: i. tokenization and ii. dilation and downsampling. Tokenization is a necessary step in masked language modeling (MLM) with bidirectional Transformer architectures (BERT) [13], a common model in genomics. These tokenizers use fixed k-mers (short overlapping sequences of length k) or frequency-based byte pair encoding (BPE), that attempt to aggregate DNA into meaningful units [25, 55]. Consequently, these aggregation techniques create large new vocabularies (compared to the natural vocabulary of 4 nucleotides) that are less generalizable [49]. The second strategy uses dilated convolutions and downsampling, both of which essentially average or skip elements between weights [18]. A canonical example is the Enformer, which uses dilation and downsampling to reach context lengths of 100k nucleotides to predict gene expression tracks [1]. Common across tokenization, dilation, and downsampling is the sacrifice of single nucleotide resolution to reach longer context.

### 2.3    Large Convolutional Models

A discrete convolution between an input $x$ of length $L$ and a (learnable) filter $h$ is given by:

$$y_t = (h * x)_t = \sum_{t'=0}^{L-1} h_{t-t'} x_{t'} \quad \text{or equivalently} \quad y = \mathsf{T}x. \tag{2.2}$$

where $\mathsf{T} \in \mathbb{R}^{L \times L}$ is the Toeplitz matrix corresponding to the convolution. Historically, convolutions have played an important role in deep learning and more broadly signal processing. More recently, it has been shown that by stacking $k$ long convolution layers, where $k$ is parametrized through a function $\gamma_\theta$ i.e. $k := \gamma_\theta(L)$, one can achieve state-of-the-art performance on a variety of benchmarks involving long sequences, for example the Long Range Arena (LRA) [48, 24, 47, 19]. Different $\gamma_\theta$ have been proposed in the literature: state-space models [24, 19], and implicit parametrizations via neural fields [45, 44, 37]. On language, the H-family of implicit convolution language models,

H3 and Hyena, [12, 37] used long convolutions and gating to match Transformer performance in $\mathcal{O}(L \log_2 L)$ time, notably lower than the $\mathcal{O}(L^2)$ of attention-based models.

HyenaDNA takes inspiration from these approaches, showing that attention-free, long-context causal models can achieve high performance on downstream genomic tasks. These extended long-range capabilities enable us to explore new paradigms in genomics, such as in-context learning to easily adapt to new tasks without updating pretrained models.

# 3 HyenaDNA Long-Range Genomic Foundation Models

In this section, we introduce the HyenaDNA approach to long-range genomic sequence modeling. We start with a description of the model architecture, then discuss sequence length warm-up and soft prompting techniques for downstream adaptation.

## 3.1 The HyenaDNA Model

The HyenaDNA model is a decoder-only, sequence-to-sequence architecture defined by a stack of blocks consisting of a Hyena operator [37], followed by a feed-forward neural network (see Fig. 1.3).

Given an input $x \in \mathbb{R}^L$ ($L$ denotes sequence length), a Hyena[3] operator can be defined as:

$$(x_1, x_2, v) \mapsto \mathsf{H}(x_1, x_2)v$$
$$\mathsf{H}(x_1, x_2) = \mathsf{D}_{x_2} \mathsf{T}_h \mathsf{D}_{x_1} \qquad (3.1)$$

where $x_1$, $x_2$, $v$ are projections of the input, and $\mathsf{T}_h \in \mathbb{R}^{L \times L}$ is the Toeplitz matrix constructed from a learnable long convolution filter produced as the output of a neural network, $(\mathsf{T}_h)_{ij} = h_{i-j}$. The convolution filter values themselves are obtained through a small neural network $\gamma_\theta$ taking as input

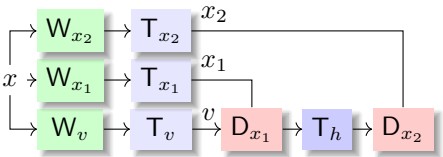

Figure 3.1: The Hyena operator is a combination of long convolutions T and data-controlled gating D, and can be a drop-in replacement for attention.

the time (position) index and optionally positional encodings, $h_t = \gamma_\theta(t)$, which enable the operator to process very long sequences without growing linearly in the number of parameters. Further, the matrices $\mathsf{D}_{x_1}, \mathsf{D}_{x_2} \in \mathbb{R}^{L \times L}$ are constructed with $x_1, x_2$ on the diagonals, and evaluated as element-wise gating. The projections are obtained by applying a dense linear layer and short convolution to the input sequence, as shown in Figure 3.1.

**Proposition 3.1.** *A* Hyena *operator can be evaluated in* $\mathcal{O}(L \log_2 L)$ *time.*

Efficient evaluation is crucial on settings involving extremely long sequences such as genomics. In the general case where the embedding dimension $D > 1$ and $x \in \mathbb{R}^{L \times D}$, the linear projections $\mathsf{W}_{x_1}, \mathsf{W}_{x_2}, \mathsf{W}_v \in \mathbb{R}^{D \times D}$ are right multiplied to $x$, and $D$ independent Hyena operators are then applied to each dimension.

## 3.2 Training Long Sequence Models

**Tokenization**   The subquadratic cost of HyenaDNA in sequence length allows the model to process ultralong sequences directly at the single nucleotide level without the need for frequency-based aggregation tokenizers. This enables fine-grain resolution for both short and long sequences, critical for detecting single nucleotide polymorphisms or mutations and modeling long-range dependencies in gene expression.

We use the natural DNA vocabulary and refer to each nucleotide as a token. The tokens include "A", "G", "C", "T", and "N" (a non-specific nucleotide) and special character tokens for padding, separation, and unknown characters. Tokens are mapped to embedding dimension $D$.

**Sequence length warm-up for ultralong sequences**   Directly training on long sequences can affect training stability as the variance in gradient increases [28]. Training on shorter sequences initially (followed by longer sequences) was used by [38] to train small scale Transformers and reduce training time, while [28] used sequence length warm-up to address stability on up to 2k tokens.

---

[3]We discuss $D = 1$ and order 2 Hyena operators for simplicity.

For ultralong sequences (200k+), we develop a new warm-up schedule that gradually increases the sequence length in stages to improve both stability and decrease training time.

Our sequence length schedule starts at $L_1 = 64$, then doubles the window at each stage while keeping the global batch size constant. By doing so, iterations at each consecutive stage will include more tokens, ensuring the scheduler can also act as a form of batch size warm-up. In Fig. 3.2, we observe sequence length scheduling to be particularly important at sequence lengths greater than 450k, where at this length training time is reduced by 40% and improving ultimate accuracy by 7.5% points for a species classification task described later in section 4.4.3.

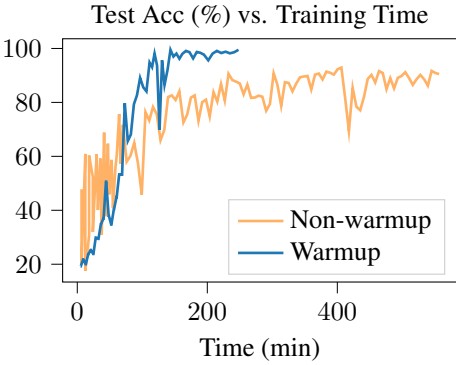

Figure 3.2: Sequence length warm-up reduces the training time of HyenaDNA at sequence length 450k by 40% and boosts accuracy by 7.5 points on species classification.

### 3.3 Downstream Adaptation

**Tuneable prompting for long-context models**
Prompts have been traditionally used to guide the output of a FM [30] by prepending additional context to an input. Expanding on this approach, *soft* tuneable prompting was introduced to inject *learnable* tokens (as weights) into the input directly [27] as an alternative to model fine-tuning.

With an extended context length ($L$), we're able to explore new paradigms in adapting FMs after pretraining. Given a downstream task with prompts $x_p \in \mathbb{R}^T$ and corresponding labels $y_p$, we prepend $N \leq L - T$ trainable parameters $\theta$ of dimension $D$ after the embedding step:

$$x \leftarrow \texttt{concat}[\texttt{embed}(x_p), \theta], \quad x \in \mathbb{R}^{L \times (T+N)} \tag{3.2}$$

The resulting sequences $x$ are then processed by the model, and $\theta$ is optimized on a loss function involving the input sequence's label $y_p$. Crucially, soft prompting requires utilization of a small subset of prompt and label pairs to optimize $\theta$.

During soft prompting, HyenaDNA only optimizes the parameters of the prompt in the input sequence while keeping all other model parameters fixed. Soft prompting thereby provides a flexible and computationally efficient approach to adapting genomic FMs to new downstream tasks.

## 4 Experiments

In 4.1, we start with pretraining HyenaDNA on the human reference genome [22]. We then evaluate HyenaDNA on existing short-range (<5k nucleotides) downstream benchmarks in 4.2 to assess the performance of single nucleotide resolution. In 4.3, we explore what new capabilities emerge with longer range genomic modeling in the form of in-context learning. Finally, we push the limits of ultralong context performance in 4.4.

### 4.1 Pretraining on the Human Genome

We pretrain HyenaDNA on the human reference genome [22] using next nucleotide (token) prediction. Starting with a stack of decoder-only Transformer blocks, we swap attention for the Hyena operator, and compare against a baseline Transformer (GPT) with Flash Attention [11]. We add gradient checkpointing to HyenaDNA to decrease the memory footprint by 3x on longer sequences ( > 160k). We then scale HyenaDNA along dimensions of model depth (2 to 8 layers), width (128 to 256 dimensions), and sequence length (1024 to 1M). At sequence length 1M, HyenaDNA is 160x faster than its Transformer counterpart as shown in Fig. 4.1.

As shown in Fig. 1.2, we observe that as context length increases, perplexity improves during pretraining. However, this improvement comes at the expense of more training time and tokens. For models too shallow to effectively process longer context, perplexity can begin to degrade (increase), observing inflection points with longer sequences. In this way, increasing context can serve as a

novel regularization dimension. For genomic pretraining, we provide the following guidelines. 1. In optimizing for faster training time, shorter context enable lower perplexity to be reached faster. 2. In optimizing for best overall perplexity, longer context allows for lower perplexity at the cost of training on more tokens. See A.1 for experiment details.

## 4.2 Single Nucleotide Resolution

Our first downstream tasks use short-range genomic sequences (<5k) aimed at evaluating single nucleotide resolution performance on sequence-level classification using standard fine-tuning.

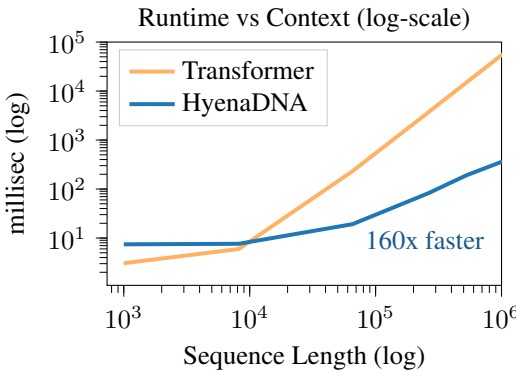

Figure 4.1: Runtime (forward & backward pass) for Transformer & HyenaDNA: 2 layers, width=128, gradient checkpoint, batch size=1, A100 80GB. At 1M tokens HyenaDNA is **160x faster** than Transformer.

**GenomicBenchmarks** We start with the newly released GenomicBenchmarks [23], which is comprised of 8 regulatory element classification datasets with sequence lengths of 200-500, and one up to 4,776. The original baseline model uses a short-range CNN. We fine-tune the pretrained Transformer (GPT) and HyenaDNA from 4.1, both having single nucleotide resolution, as well as the DNABERT model [25]. HyenaDNA sets a new SotA on 7 of 8 datasets and by up to 20% points on the human enhancer identification task, as shown in Tab. 4.1. See A.2 for additional experiment details and ablations.

**Nucleotide Transformer** Next, we benchmark against 18 datasets from the Nucleotide Transformer (NT) [10], which includes predicting regulatory elements for enhancers, promoters, epigenetic marks, and splice sites from DNA sequences of length 200-600 nucleotides. We compare against 3 NT base models, which were pretrained using masked language

Table 4.1: **GenomicBenchmarks** Top-1 accuracy (%) for pretrained HyenaDNA, DNABERT and Transformer (GPT from 4.1), and the previous SotA baseline CNN (scratch).

| DATASET | CNN | DNABERT | GPT | HYENADNA |
|---|---|---|---|---|
| Mouse Enhancers | 69.0 | 66.9 | 80.1 | **85.1** |
| Coding vs Intergenomic | 87.6 | **92.5** | 88.8 | 91.3 |
| Human vs Worm | 93.0 | 96.5 | 95.6 | **96.6** |
| Human Enhancers Cohn | 69.5 | 74.0 | 70.5 | **74.2** |
| Human Enhancers Ensembl | 68.9 | 85.7 | 83.5 | **89.2** |
| Human Regulatory | 93.3 | 88.1 | 91.5 | **93.8** |
| Human Nontata Promoters | 84.6 | 85.6 | 87.7 | **96.6** |
| Human OCR Ensembl | 68.0 | 75.1 | 73.0 | **80.9** |

modeling (BERT) and then fine-tuned. The NT models ranged from 500M to 2.5B parameters, and pretrained on up to 3202 genomes. All NT models use 6-mer sequences of 1000 tokens long. For HyenaDNA, we attach a linear decoder head and fine-tune a pretrained model, surpassing SotA on 12 of 18 datasets using a model with orders of magnitude less parameters and pretraining data, shown in Tab. 4.2. See A.2 for additional experiment details and ablations.

## 4.3 In-context Learning for Genomic Sequences

Compared to natural language FMs, which have shown strong success with in-context learning, HyenaDNA's vocabulary is very small. DNA sequences are also less diverse in structure, e.g. there's no concept of labels or descriptions that follow a DNA sequence. This makes it challenging to perform "pure" in-context learning (relying only on inference), since new concepts such as classification labels would require new symbols. To overcome this limitation and explore the potential for in-context learning in genomics, we make use of two variants of in-context learning: soft prompting and instruction fine-tuning. Each involve a brief tuning phase to introduce the concept of classification using only the existing vocabulary.

**Procedure** In both variants, we use the GenomicBenchmarks in 4.2, and a HyenaDNA model pretrained on sequence length 160k from 4.1.

In the first experiment, we evaluate a soft prompting approach by prepending a sequence of soft tuneable tokens (2 to 32k) directly in the input sequences. We include a brief tuning phase ($< 20$ epochs), updating the soft tokens only, to provide HyenaDNA with the ability to indicate the target classes. To denote classes, we repurpose HyenaDNA's fixed vocabulary: for binary classification, for example, we indicate the two classes with the letters "A" and "N".

In the second experiment, we evaluate a few-shot learning approach to in-context learning [6] by prepending, consecutively, $k$ (2 to 32) demonstrations of each class and its sequence into the prompt. As before, we encode class labels by the use of individual letters of HyenaDNA's existing vocabulary. We additionally perform a brief instruction-tuning period [52] for each dataset to familiarize HyenaDNA with this task structure by tuning the pretrained model on a small subset of the dataset.

Table 4.2: **Nucleotide Transformer (NT) Benchmarks** The Matthews correlation coefficient (MCC) is used as the performance metric for the enhancer and epigenetic marks dataset, and the F1-score is used for the promoter and splice site dataset.

| MODEL | NT | NT | NT | HyenaDNA |
|---|---|---|---|---|
| PARAMS | 500M | 2.5B | 2.5B | 1.6M |
| # OF GENOMES | 1 | 3,202 | 850 | 1 |
| Enhancer | 53.5 | 59.3 | 58.0 | **62.6** |
| Enhancer types | 48.5 | 50.0 | 47.4 | **55.7** |
| H3 | 73.7 | 77.6 | 81.4 | **81.7** |
| H3K4me1 | 35.8 | 44.5 | 55.9 | **57.1** |
| H3K4me2 | 28.1 | 30.0 | 32.6 | **53.9** |
| H3K4me3 | 26.3 | 28.1 | 42.1 | **61.2** |
| H3K9ac | 46.2 | 50.8 | 57.5 | **65.1** |
| H3K14ac | 37.7 | 47.1 | 55.0 | **66.3** |
| H3K36me3 | 46.7 | 53.3 | 63.2 | **65.3** |
| H3K79me3 | 57.7 | 59.2 | 64.2 | **71.6** |
| H4 | 76.2 | 78.9 | **82.2** | 79.6 |
| H4ac | 34.4 | 42.3 | 50.1 | **63.7** |
| Promoter all | 95.4 | 96.6 | **97.4** | 96.5 |
| Promoter non-TATA | 95.6 | 96.9 | **97.7** | 96.6 |
| Promoter TATA | 94.8 | 95.8 | 96.4 | **96.7** |
| Splice acceptor | 96.5 | 98.5 | **99.0** | 96.6 |
| Splice donor | 97.2 | 98.2 | **98.4** | 97.3 |
| Splice all | 97.2 | 97.8 | **98.3** | 97.9 |

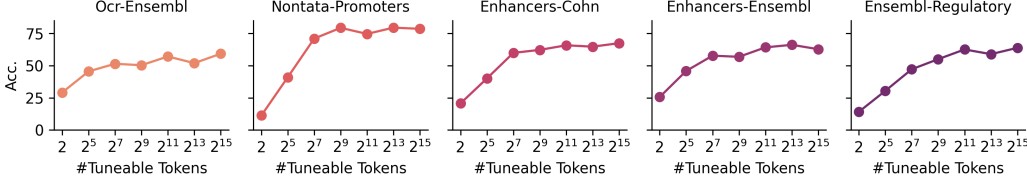

Figure 4.2: **Filling long-context with soft tuneable tokens.** HyenaDNA is able to learn new tasks in-context when adding a sequence of tuneable tokens to the input sequences. Longer sequences of tuneable tokens lead to better performance.

**Results** In Fig. 4.2, HyenaDNA's performance on novel tasks improves as more tuneable tokens are added into the input sequences, and saturates close to baseline performance (Tab. 4.1; with the exception of the Human Regulatory dataset). By contrast, we find that increasing $k$-shot demonstrations to the input does not necessarily improve performance. A higher number of tuning samples is needed before $k$-shot demonstrations start to boost accuracy as shown in Tab. A.1. See A.3 for experiment details.

## 4.4 Ultralong-Range Genomics

In our final experimental section, we focus on pushing the limits of using long context effectively in genomics. In 4.4.1, we tackle a challenging 919 binary multi-task against a sparse-attention baseline. In 4.4.2 we analyze the learned embeddings HyenaDNA and its use in clustering long sequences by functional annotation, and in 4.4.3 we showcase a novel ultralong-range species classification task.

### 4.4.1 Chromatin Profile Prediction

The prediction of chromatin profiles and epigenetic markers from DNA sequences is an important and challenging task to quantify the functional effects of non-coding variants. These variants include single nucleotide changes in DNA that can affect the downstream expression of genes [56]. The DeepSEA dataset [57] is compiled from 919 chromatin features including transcription factor (TF) binding profiles, DNase I-hypersensitive sites (DHS) and histone mark (HM) profiles. For a given sequence, the task is to jointly predict 919 labels cor-

Table 4.3: **Chromatin profile prediction** Median AUROC computed over three categories: Transcription factor binding profiles (TF), DNase I-hypersensitive sites (DHS) and histone marks (HM).

| MODEL | PARAMS | LEN | AUROC | | |
|---|---|---|---|---|---|
| | | | TF | DHS | HM |
| DeepSEA | 40 M | 1k | 95.8 | 92.3 | 85.6 |
| BigBird | 110 M | 8k | 96.1 | 92.1 | 88.7 |
| HyenaDNA | 7 M | 1k | **96.4** | **93.0** | 86.3 |
| | 3.5 M | 8k | 95.5 | 91.7 | **89.3** |

responding to the chromatin profile (similar to peak detection) of a central region of the sequence, indicating the presence of such functional effects. The input also includes flanking regions that provide broader contextual information needed to incorporate long-range interactions. We fine-tune our pretrained HyenaDNA models from 4.1 and perform competitively against a DeepSea CNN and the SotA sparse attention BigBird [55] baselines using 5-30× fewer parameters. See A.4 for experiment details.

### 4.4.2 Biotype Embeddings

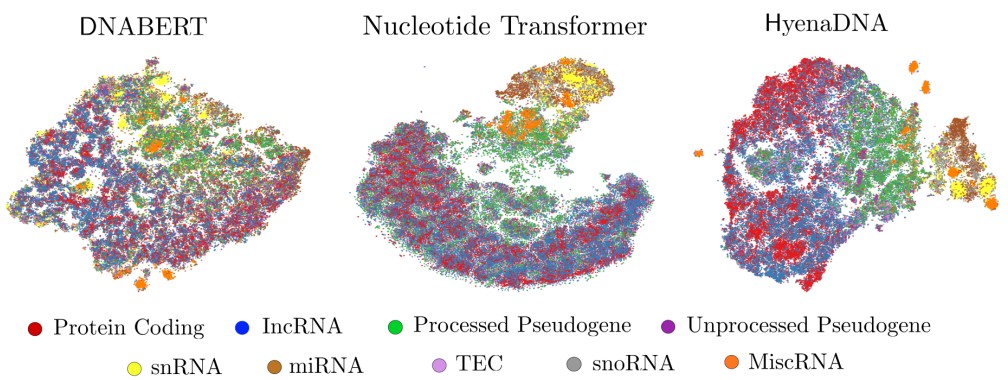

## Sequence embeddings, colored by biotype

DNABERT          Nucleotide Transformer          HyenaDNA

● Protein Coding  ● lncRNA  ● Processed Pseudogene  ● Unprocessed Pseudogene
● snRNA  ● miRNA  ● TEC  ● snoRNA  ● MiscRNA

Figure 4.3: **Embedding visualisation.** t-SNE of the embeddings generated by DNABERT, Nucleotide Transformer and HyenaDNA coloured by Ensembl biotype annotations.

Next, we analyze the pretrained embeddings from HyenaDNA and compare them with DNABERT [25] and the Nucleotide Transformer [10]. We encode sequences of human genes corresponding to different biological function annotations obtained from the Ensembl dataset known as biotypes [9]. In cases where the length of the input exceeds the context window of the encoder, the sequence is chunked (by the max length of the encoder) and averaged.

We fit the embeddings using an XGBoost [7] classifier on the 10 most frequent biotypes, and apply t-SNE [50] for visualization. As shown in 4.3, distinct clusterings emerge visually, while quantitatively, HyenaDNA produces the highest F1 score in biotype classification (with a much smaller model), indicating that during pretraining, HyenaDNA learns informative features related to biological function.

Table 4.4: **Embedding quality** Weighted F1 classification score on 10 biotypes.

| MODEL | PARAMS | LEN | F1 |
|---|---|---|---|
| DNABERT | 110 M | 512 | 64.6 |
| NT | 500 M | 6k | 66.5 |
| HyenaDNA | 7 M | 160k | **72.0** |

### 4.4.3 Species Classification

The majority of the genome is conserved across species – humans and non-human primates, for example, have <10% sequence divergence [43], making them difficult to discriminate. This allows us to to design an ultralong-range sequence modeling task to test whether a model can determine the source species of a random genetic sequence. To train, we randomly sample DNA sequences from 5 different species, and fine-tune pretrained HyenaDNA and Transformer models from 4.1 to predict the species label. We observe in Tab. 4.5 that both models struggle on shorter sequences of length 1024, but performance improves with longer contexts as the distinct mutational profile of each species becomes more evident. HyenaDNA effectively solves the task by using a context length of 450k to 1 million, where Transformer cannot due to infeasible training time limitations. See A.6 for experiment details.

## 5 Conclusion

**Summary** We presented HyenaDNA, a genomic foundation model pretrained on the human reference genome with context lengths up to 1 million tokens at single nucleotide resolution - an up to 500x increase over previous genomic FMs using dense-attention. HyenaDNA is able to learn generalizable features that can then be fine-tuned for tasks including identifying regulatory elements and on a 919-way chromatin profile prediction task. We also explored the first use of in-context learning in genomics to enable simpler adaptation to downstream tasks without any updates to pretrained weights.

**Limitations and Future Work** While demonstrating competitive results and introducing novel capabilities, it is worth noting that HyenaDNA was pretrained on only one human reference genome. Incorporating genomes of multiple humans and species could increase generalizability in learned features and reduce bias. Furthermore, our current focus in this study was exclusively on DNA sequences. Extending our framework to incorporate other

Table 4.5: **Species classification** Top-1 accuracy (%) for 5-way classification (human, lemur, mouse, pig, hippo). The ✗ symbol indicates infeasible training time.

| Model | Len | Acc |
|---|---|---|
| Transformer | 1k | 55.4 |
| HyenaDNA | 1k | 61.1 |
| Transformer | 32k | 88.9 |
| HyenaDNA | 32k | 93.4 |
| Transformer | 250k | ✗ |
| HyenaDNA | 250k | 97.9 |
| Transformer | 450k | ✗ |
| HyenaDNA | 450k | 99.4 |
| Transformer | 1M | ✗ |
| HyenaDNA | 1M | **99.5** |

biological or chemical sequences, such as proteins and drug molecules, has the potential to unlock multi-modal capabilities similar to those observed in natural language and vision FMs [39, 40, 54].

With respect to model size, HyenaDNA is significantly smaller than previous genomic FMs and was pretrained using up to 8 Nvidia A100 (80GB) GPUs. We expect increasing model size, and compute, may lead to additional long-range capabilities. Notably, with model parallelism, it becomes feasible to extend the context length by orders of magnitude beyond this current work, and leave that open to future research.

Furthermore, beyond discriminative applications, the use of long context models in generative tasks unlocks exciting prospects for the design of synthetic regulatory elements, genes and protein complexes. In conclusion, the continued advancements of long-range sequence models with single nucleotide resolution hold great promise in driving innovation in genomic research and unraveling the complexities of biological systems.

## Acknowledgments

We would like to thank Guatam Machiraju, Elliott Epstein, Archis Joglekar, Jared Dunnmon, Nazim Bouatta and Anshul Kundaje for helpful discussion and feedback on earlier drafts, and Together for providing the compute used to train models in this paper. We gratefully acknowledge the support of NIH under No. U54EB020405 (Mobilize), NSF under Nos. CCF1763315 (Beyond Sparsity), CCF1563078 (Volume to Velocity), and 1937301 (RTML); US DEVCOM ARL under No. W911NF-21-2-0251 (Interactive Human-AI Teaming); ONR under No. N000141712266 (Unifying Weak Supervision); ONR N00014-20-1-2480: Understanding and Applying Non-Euclidean Geom-

etry in Machine Learning; N000142012275 (NEPTUNE); NXP, Xilinx, LETI-CEA, Intel, IBM, Microsoft, NEC, Toshiba, TSMC, ARM, Hitachi, BASF, Accenture, Ericsson, Qualcomm, Analog Devices, Google Cloud, Salesforce, Total, the HAI-GCP Cloud Credits for Research program, the Stanford Data Science Initiative (SDSI), Department of Defense (DoD) through the National Defense Science and Engineering Graduate Fellowship (NDSEG) Program, and members of the Stanford DAWN project: Facebook, Google, and VMWare. This work is supported by NSF (1651565), AFOSR (FA95501910024), ARO (W911NF-21-1-0125), ONR, DOE (DE-SC0022222), CZ Biohub, and Sloan Fellowship. The U.S. Government is authorized to reproduce and distribute reprints for Governmental purposes notwithstanding any copyright notation thereon. Any opinions, findings, and conclusions or recommendations expressed in this material are those of the authors and do not necessarily reflect the views, policies, or endorsements, either expressed or implied, of NIH, ONR, or the U.S. Government.

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

# HyenaDNA

## *Supplementary Material*

## Contents

# A Appendix: Experimental Details

In the following sections we provide further details for each experiment. Across all experiments, we use Pytorch and Pytorch Lightning. We train on a mix of Nvidia GPUs with A100s, V100s, and T4s. Unless otherwise stated, we use a cross entropy loss for our objective. Our repository is made public here: https://github.com/HazyResearch/hyena-dna.

## A.1 Pretraining Details

Table A.1: Hyperparameter settings for HyenaDNA pretraining (select models).

| Layers | 2 | 2 | 4 | 4 | 8 |
|---|---|---|---|---|---|
| Width | 128 | 256 | 128 | 256 | 256 |
| Params (M) | 0.44 | 1.6 | 0.87 | 3.3 | 6.6 |
| Max seq. len. | 64k | 64k | 64k | 64k | 1M |
| Optimizer | AdamW | | | | |
| Optimizer momentum | $\beta_1, \beta_2 = 0.9, 0.999$ | | | | |
| Learning rate | 1.5 - 6e-4 | | | | |
| LR Scheduler | Cosine decay | | | | |
| Batch size | 64 - 256 | | | | |
| Global steps | 10 - 20k | | | | |
| Weight decay (model) | 0.1 | | | | |
| Weight decay (Hyena layers) | 0 | | | | |
| Embed dropout | 0.1 | | | | |
| Residual dropout | 0 | | | | |

**Data**   For pretraining, we use a single human reference genome [22], and leverage the training and validation intervals (start and end) from [1]. During training, we sample an interval and obtain a sequence of length $L$ by adjusting the intervals on both ends. For the test set, we use chromosomes 14 and X, exclusively, and sample non-overlapping sequences of length $L$.

**Model**   We design a suite of parameter efficient architectures with depths between 2 and 8 layers, Hyena blocks of Order-N = 2, and width 128 to 256. The MLP expansion factor (reverse bottleneck) is 4x the width. See Fig. 1.3 for the block architecture of HyenaDNA. The parameter counts range from 400k to 6.6M, trained on sequence lengths between 1,024 and 1M. Tab. A.1 highlights a representative subset of the models we trained. Note: we use different pretrained model sizes depending on the downstream task to prevent overfitting. When selecting which pretrained model to use for a downstream task, we found that a pretrained sequence length of 2 to 4x the downstream max sequence length results in the best performance.

**Training**   We pretrain each model for 10-20k global steps. For models trained on longer sequences, this translates to more tokens being used, as each sample contains more tokens. For example, the largest model with context length 1M was trained on 2T tokens over 4 weeks. We adjust the "accumulate_grad_batches" argument in Pytorch Lightning to keep the global batch size consistent across models and sequence lengths. See Tab. A.1 for hyperparameter details.

**Training efficiency**   We compare pretraining compute resources and GPU-hours to reach competitive performance on the short-range tasks for several baselines and HyenaDNA models, shown in Tab. A.2.

Table A.2: Pretraining GPU & runtime comparison for short-range models.

| | DNABERT | NUCLEOTIDE TRANSFORMER | HyenaDNA | HyenaDNA |
|---|---|---|---|---|
| Params | 110M | 2.5B | 436K | 1.6M |
| GPUs | 8-2080 TI | 128-A100-80GB | 1-A100-40GB | 1-A100-40GB |
| Wall clock | 25 days | 28 days | 80 mins | 80 mins |
| GPU-hrs | 12,000 | 215,000 | 1.3 | 1.3 |

### A.2 Short-Range Genomics Details

#### A.2.1 GenomicBenchmarks experiment

**Data**   The GenomicBenchmarks [23] includes 8 datasets designed for sequence-level classification tasks that involve predicting regulatory elements, along with one binary species task. The benchmarks provided for the baseline model include two sets of results: one obtained with Pytorch and the other with TensorFlow. Since our code base is implemented in Pytorch, we compare our results with the Pytorch-based benchmarks.

**Model**   Our backbone is a pretrained 2 layer HyenaDNA model with width 128, trained on sequence length 1024. We pool along the sequence dimension to obtain a classification token, and attach a simple linear decoder head. The baseline CNN, as described by [23], uses uses an embedding layer, 3 convolutional layers with number of filters: 16, 8, and 4. It uses batch norm and max pooling after each convolutional layer, followed by 2 dense layers. It is trained for 10 epochs with batch size 64. The mode sizes range from 120k to 520k, depending on sequence length chosen.

Table A.3: GenomicBenchmarks hyperparameters for HyenaDNA and the baseline Transformer (GPT from 4.1), which uses FlashAttention [11].

|  | TRANSFORMER | HyenaDNA |
| --- | --- | --- |
| Layers | 2 | 2 |
| Width | 128 | 128 |
| Parameters | 529k | 436k |
| Learning rate | $1\text{-}6e^{-4}$ | $1\text{-}6e^{-4}$ |
| Weight decay (model) | 0-0.2 | 0-0.2 |
| Weight decay (Hyena layers) | - | 0 |
| Embed dropout | 0-0.2 | 0.0-0.3 |
| Resid dropout | 0-0.2 | 0-0.3 |
| Num heads | 8 | - |
| Optimizer | AdamW | |
| Optimizer momentum | $\beta_1, \beta_2 = 0.9, 0.999$ | |
| LR scheduler | Cosine decay | |
| Batch size | 128-1024 | |
| Training epoch | 100 | |
| Reverse complement aug. | true/false | |
| Sequence lengths | 200-4800 | |

**Training**   The primary hyperparameters we sweep across include: learning rate, global batch size, dropout, weight decay, and a reverse complement augmentation. See Tab. A.3 for ranges of hyperparamters used.

#### A.2.2 Ablations on the GenomicBenchmarks

To better understand how specific design choices in the HyenaDNA model effect performance, we perform a series of ablation experiments on the GenomicBenchmarks.

**Pretraining:**   We train HyenaDNA from scratch and compare with the pretrained version. The pretrained models provide mild to moderate gains - likely due to the benchmarks being near saturation already.

**Tokenization:**   We train HyenaDNA using a k-mer tokenizer (k=6) to isolate the effect of the single nucleotide tokenizer. The k-mer tokenizer drops performance significantly across on a majority of the datasets (by as much as 10 accuracy points), while boosting one dataset (Human Enhancer Ensembl). Therefore, the single nucleotide tokenization appears to be a significant component of the HyenaDNA model.

**Bidirectional:**   To ablate the impact of using a causal model, we implemented a bidirectional version of HyenaDNA and trained from scratch on the GenomicBenchmarks (i.e. without masked

Table A.4: **GenomicBenchmarks Top-1 accuracy (%)** GPT is the causal Transformer from 4.1, HyenaDNA k-mer uses a 6-mer tokenizer, and HyenaDNA bidirection is a bidirectional version of the Hyena operator.

| MODEL | GPT | GPT | HyenaDNA | HyenaDNA | HyenaDNA k-mer | HyenaDNA bidirection | DNABERT |
|---|---|---|---|---|---|---|---|
| Pretrained | no | yes | no | yes | no | no | yes |
| Mouse Enhancers | 79.3 | 79.3 | 84.7 | **85.1** | 81.8 | 80.6 | 66.9 |
| Coding vs Intergenomic | 89.3 | 91.2 | 90.9 | 91.3 | 86.7 | 90.3 | **92.5** |
| Human vs Worm | 94.8 | **96.6** | 96.4 | **96.6** | 92.9 | 95.9 | 96.5 |
| Human Enhancers Cohn | 67.7 | 72.9 | 72.9 | **74.2** | 69.8 | 72.1 | 74.0 |
| Human Enhancers Ensembl | 79.0 | 88.3 | 85.7 | **89.2** | 88.0 | 85.9 | 85.7 |
| Human Regulatory | 90.2 | 91.8 | 90.4 | **93.8** | 90.2 | 89.1 | 88.1 |
| Human Nontata Promoters | 85.2 | 90.1 | 93.3 | **96.6** | 83.5 | 88.5 | 85.6 |
| Human OCR Ensembl | 68.3 | 79.9 | 78.8 | **80.9** | 70.2 | 75.3 | 75.1 |

language model pretraining). The bidirectional version degraded performance on 7 of 8 datasets compared to the standard causal HyenaDNA (also from scratch), on average by 3.8 accuracy points.

The bidirectional HyenaDNA was implemented by using a circular FFT convolution. This involved manipulating the padding on the input sequence before performing the FFT convolution. Previously, we zero padded the input on the right side by length $L$ (the sequence length). For bidirectionality, we pad by $1/2\ L$ on the left and right side of the input, effectively providing a bidirectional receptive field (due to the circular convolution). This is one of many possible ways to implement a bidirectional version of Hyena.

### A.2.3 Downstream prediction tasks for Nucleotide Transformer benchmark

Following the Nucleotide Transformer [10], we collected datasets from four different sources [21, 35, 34, 46].

**Promoter** The promoter dataset included TATA-box-containing and TATA-box-lacking promoters. Tasks involved predicting promoters with a TATA-box, identifying promoters lacking a TATA-box, and distinguishing between both promoter categories and non-promoter sequences. The promoter datasets were obtained from the Eukaryotic Promoter Database (EPDnew)[4] for human and mouse genomes. Promoter sequences were extracted from regions 249 nucleotides upstream and 50 nucleotides downstream of the transcription start sites.

**Enhancer** For the enhancer prediction task, we used the dataset from [21] containing DNA sequences classified into strong enhancers, weak enhancers, and non-enhancers. The tasks involved binary classification to distinguish enhancer sequences from non-enhancer sequences and identify specific enhancer types.

**Epigenetic Marks** In the epigenetic marks prediction task, we used the dataset from [35, 36] to predict nucleosome occupancy and modification states in the yeast genome. In 10 binary classification tasks, the model had to discriminate between DNA regions that were occupied by histones or not. The 10 tasks varied based on the types of histones investigated, including unmodified histones H3 and H4, as well as histones modified by either acetylation (H3K9ac, H3K14ac) or methylation (H3K4me1, H3K4me2, H3K4me3, H3K36me3, H3K79me3).

**Splice Site** For the splice site prediction task, DNA sequences from over 100 organisms were used to predict whether the sequences contain donor or acceptor splice sites [46]. Donor splice sites denote the beginning of an intron and acceptor splice sites the end of an intron. During RNA splicing, these sites are recognized by the spliceosome, a complex molecular machine that enables the removal of introns from the gene.

---

[4]https://epd.epfl.ch//index.php

Table A.5: Hyperparameter ranges used to fine-tune HyenaDNA for all Nucleotide transformer datasets. Exact hyperparameters per dataset can be found in our code repository.

|  | HyenaDNA |
| --- | --- |
| Layers | 2 |
| Width | 256 |
| Parameters | 1.6M |
| Optimizer | AdamW |
| Optimizer momentum | $\beta_1, \beta_2 = 0.9, 0.999$ |
| Training epoch | 100 |
| Batch size | 256-1024 |
| Learning rate | 2e-4 to 1e-3 |
| LR scheduler | Cosine decay |
| Weight decay (model) | 0-0.2 |
| Weight decay (Hyena layers) | 0 |
| Embed dropout | 0-0.2 |
| Resid dropout | 0-0.2 |
| Reverse complement aug. | true/false |
| Sequence lengths | 200-600 |

**Preprocessing**  The Nucleotide Transformer study did not provide their exact train-test splits, except for the enhancer dataset. Therefore, we generated our own train-test splits using a 90:10 ratio. For the promoter dataset, negative samples were not available, and had to be generated following the procedure described by [34].

**Model & Training**  For the architecture, we use a HyenaDNA model with 2 layers and width 256, and trained on sequences of length 1024. We average across the tokens to obtain a single classification token. For each task, we replaced the model head and fine-tuned the weights of the entire model (1.6M parameters). In contrast, the Nucleotide Transformer uses a parameter-efficient fine-tuning technique that introduces new weights and fine-tunes only the newly added weights, while keeping the initial model weights frozen, presumably due to its large size of 500M to 2.5B parameters. The corresponding HyenaDNA hyperparameter ranges used for training each task are reported in Table A.5.

### A.2.4  Ablations on the Nucleotide Transformer benchmarks

We perform additional ablations on the Nucleotide Transformer benchmarks to assess the impact of pretraining, as well as attention vs. HyenaDNA, as shown in shown in Table A.6. We observed that pretraining has a greater effect on the more challenging tasks (and as sequences become longer, shown in A.11). On the more challenging tasks (histone marks, datasets starting with "H"), pre-training boosts HyenaDNA metrics by up to 21 MCC points on H3K4me3. For simpler tasks (with higher baseline scores) such as the splice sites and promoter tasks, the gain was lower (0 to 1 accuracy points), as these were already near saturation in performance.

### A.3  In-Context Learning Details

**Background**  A key premise of foundation models is that they are able to learn new tasks with little to no new training data [4]. Recent advances in language modeling have demonstrated that language foundation models can often adopt the behaviors necessary to perform new tasks *in-context* [6]. Here, information about the task that is to be performed, such as examples of respective inputs and targets, are added to the input of the model. By conditioning their prediction on the provided context, language foundation models are generally able to perform the task without any changes to their parameters.

A key challenge for in-context learning with HyenaDNA is its limited vocabulary, which is composed of only a few nucleotides, and does not provide any vocabulary for novel downstream tasks, such as class labels. To explore the potential for in-context learning in genomics, we use two variants of in-context learning, both using a brief tuning phase to introduce HyenaDNA to the concept of

Table A.6: **Pretraining & Attention ablations on the Nucleotide Transformer (NT) benchmarks.** The Matthews correlation coefficient (MCC) is used as the performance metric for the enhancer and epigenetic marks dataset, and the F1-score is used for the promoter and splice site dataset.

| MODEL | NT | GPT | HyenaDNA | HyenaDNA |
|---|---|---|---|---|
| PARAMS | 2.5B | 1.6M | 1.6M | 1.6M |
| PRETRAIN | yes | yes | yes | no |
| Enhancer | 58.0 | 59.3 | **62.6** | 58.6 |
| Enhancer types | 47.4 | 51.9 | **55.7** | 48.4 |
| H3 | 81.4 | 75.8 | **81.7** | 79.9 |
| H3K4me1 | 55.9 | 38.7 | **57.1** | 43.4 |
| H3K4me2 | 32.6 | 28.8 | **53.9** | 34.5 |
| H3K4me3 | 42.1 | 28.3 | **61.2** | 40.2 |
| H3K9ac | 57.5 | 49.2 | **65.1** | 52.6 |
| H3K14ac | 55.0 | 41.6 | **66.3** | 48.0 |
| H3K36me3 | 63.2 | 47.8 | **65.3** | 53.4 |
| H3K79me3 | 64.2 | 58.9 | **71.6** | 59.7 |
| H4 | **82.2** | 77.7 | 79.6 | 79.1 |
| H4ac | 50.1 | 36.4 | **63.7** | 43.5 |
| Promoter all | **97.4** | 96.3 | 96.5 | 96.1 |
| Promoter non-TATA | **97.7** | 96.6 | 96.6 | 96.5 |
| Promoter TATA | 96.4 | 96.6 | **96.7** | 96.1 |
| Splice acceptor | **99.0** | 97.6 | 96.6 | 96.6 |
| Splice donor | **98.4** | 98.1 | 97.3 | 96.5 |
| Splice all | **98.3** | 98.0 | 97.9 | 97.3 |

classification with its existing vocabulary. As a test bed for this exploration, we use 5 datasets from the GenomicBenchmarks and a HyenaDNA pretrained on sequences of 160k length sequences.

In the first experiment, we apply a soft prompting approach [27] by adding a sequence of tuneable tokens to the input inself. In the second experiment, we explore a few-shot learning approach [6] to in-context learning by adding $k$ demonstrations (DNA sequence and its label) for each class of a dataset as input to the model. To indicate classes, we make use of HyenaDNA's existing vocabulary by indicating classes with specific nucleotides. For binary classification, we indicate classes with the nucleotides "A" and "N", while additionally utilising nucleotide "G" for three-way classification. During model tuning, we thereby optimise the same next-nucleotide prediction loss as used during pretraining. See Table A.7 for an overview of the optimisation settings.

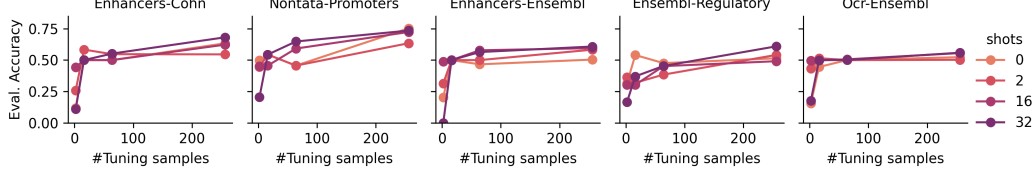

Figure A.1: **Few-shot prompting**: HyenaDNA's performance on new tasks generally improves with the number of tuning samples, but is less clear when isolating the number of $k$-shot demonstrations. With less tuning samples, the number of $k$-shot demonstrations do not improve performance. As tuning samples increase, the number of $k$-shot demonstrations start to improve performance.

**Soft prompting details** For each dataset, we prepend a sequence of $n$ (2 to 32k) learnable tokens $T_e \in \mathbb{R}^{n \times d}$, each of dimension $d$, to the input sequences $X$ of the model: $\{T_e, X, SEP\}$, where "SEP" indicates the separation token. We optimise these tuneable tokens for a maximum of 20 training epochs on the dataset's training data while keeping all other model parameters fixed. We stop training early if the model's validation loss does not improve for two epochs. After this tuning phase, we evaluate the model's performance on the dataset's full validation data. For an overview of the results of this experiment, see Fig. 4.2 of the main text.

**Few-shot prompting details**   For each dataset, we prepend a set of $k$ (0 to 32, 0 indicates regular fine-tuning) examples of each class of a dataset (so-called "shots") to an input sequence:

$$X: \quad \{X_1, \mathsf{SEP}, Y_1, \mathsf{SEP}, X_2, \mathsf{SEP}, Y_2, \mathsf{SEP}, X, \mathsf{SEP}\},$$

where $X_i$ indicates an example sequence of class $i$ with label $Y_i$ (exemplified for a two-way classification task). We tune the model on $n$ (2 to 256) such $k$-shot samples before evaluating its performance on the dataset's full validation data. For an overview of the results of this experiment, see Fig. A.1.

Table A.7: Optimization settings for in-context learning.

|  | SOFT PROMPTING | FEW-SHOT PROMPTING |
| --- | --- | --- |
| Optimizer | AdamW | AdamW |
| Optimizer momentum ($\beta_1, \beta_2$) | 0.9, 0.999 | 0.9, 0.999 |
| Learning Rate | 0.001 | 0.0001 |
| Batch Size | 16 | 2 |
| Weight Decay (model) | 0 | 0 |
| Weight Decay (Hyena layers) | 0 | 0 |
| Resid dropout | 0 | 0 |
| Embed dropout | 0.1 | 0.1 |
| Reverse complement aug. | true | false |
| LR-schedule | Plateau | - |

## A.4   Chromatin Profile Details

**Background**   Variations in non-coding regions of the genome account for the majority of disease and other trait-associated single-nucleotide polymorphisms (SNPs). For example, whilst not directly altering the sequence of an encoded protein, a SNP in a non-coding region can affect the expression of downstream genes by inducing a change in the epigenetic state [56]. Therefore predicting epigenetic markers from a given sequence is an important task in the context of quantifying the functional effects of non-coding variants. Previously DeepSEA [57], a deep convolutional sequence model, has been introduced to predict chromatin features directly from non-coding sequences.

**Data**   The authors of DeepSEA [57] compiled a dataset of 919 chromatin features from [15] and [42] including 690 TF binding profiles for 160 different TFs, 125 DHS and 104 HM profiles. The original DeepSEA dataset consists of 1000 base pair (bp) sequences from the hg19 human reference genome [8] with corresponding 919-dimension multi-label target vectors. Each label corresponds to the presence/absence of a peak in a given chromatin feature within the central 200 bp region of the sequence. The 400 bp flanking regions of the sequence provide broader contextual information which is beneficial to the task. Training and testing sets are split by chromosome and are strictly non-overlapping. In total, there are 2.2 M training samples and 227,512 samples from chromosomes 8 and 9 are held-out for testing. We use the DeepSEA chromatin profile prediction task to evaluate HyenaDNA models with varying context window. We use LiftOver [26] to convert the original DeepSEA dataset to hg38 coordinates and expand flanking regions about the central 200 bp bin symmetrically up to 8000 bp. Approximately 0.5% of samples are filtered in cases where LiftOver fails or the resulting translated sequence has a different length.

**Model**   We fine-tune several models consisting of a pretrained HyenaDNA encoder, a sequence-level pooling layer and a fully-connected decoder to perform multilabel sequence classification. We compare HyenaDNA against benchmarks set by DeepSEA, a convolutional sequence model, and BigBird [55], a sparse attention based language model. The authors of BigBird fine-tune on the DeepSEA dataset with input sequences extended to 8000 bp (asymmetrically about the center-point by -5000 and +3000 bp). Notably BigBird utilizes a byte-pair encoding tokenization scheme whereas HyenaDNA uses a single-character tokenizer and DeepSEA uses one-hot encodings. For the shortest range model (1k), we average across all tokens to perform sequence-level pooling. Whereas in the longer context model (8k) we find that extracting the last token in the sequence as the input to the fully-connected decoder performs better. We also find that for the longer context

model using an encoder pretrained on sequences larger than those used in fine-tuning was beneficial. The hyperparameters of the models used in these experiments are shown in Table A.8. Note that we reduced the depth and of models with increasing context window due to limitations on compute cost/time.

**Results**   The performance of the fine-tuned HyenaDNA models are summarised in Table 4.3. We find that the smallest sequence length model (1024 bp) outperforms both DeepSEA and BigBird on TF and DHS prediction. We find that the model pretrained on 32k sequences with only 4 layers and fine-tuned on 8k sequences outperforms BigBird on the long range HM task but suffers from degraded performance on the short range tasks. However, we postulate that this performance loss may be recovered by increasing the depth of the model. We also remark that our models contain 5-30$\times$ fewer parameters compared to DeepSEA and BigBird.

Table A.8: **Chromatin profile model settings.** HyenaDNA hyperparameter settings used in the chromatin profile prediction experiments (fine-tuning).

|  | HyenaDNA | |
| --- | --- | --- |
| Sequence length | 1024 | 8k |
| Context window | 1024 | 32770 |
| Width | 256 | 256 |
| Layers | 8 | 4 |
| Pooling method | Average | Last token |
| Parameters (M) | 6.6 | 3.5 |
| Optimizer | AdamW | AdamW |
| Optimizer momentum | $\beta_1, \beta2 = 0.9, 0.999$ | $\beta_1, \beta2 = 0.9, 0.999$ |
| Weight decay (model) | 0.1 | 0.1 |
| Weight decay (Hyena layers) | 0 | 0 |
| Embed dropout | 0.1 | 0.1 |
| Learning rate | 6e-4 | 6e-4 |
| Batch size | 64 | 64 |
| Epochs | 50 | 50 |

### A.5   Biotype Embeddings Analysis Details

**Background**   Sequence embeddings are useful in reducing dimensionality and capturing semantic relationships into fixed length vectors. We analyze pretrained embedding quality from HyenaDNA and show that it learns biologically informed features. We utilize linear probing, freezing the weights on a pretrained model and attaching a linear classification head to predict biotype sequences. We also use t-SNE to visualize clusterings that emerge from the embeddings.

**Data**   The Ensembl database [9] is a comprehensive resource for gene and transcript annotations such as biotypes. Ensembl biotypes are a classification system, based on a combination of experimental evidence and computational predictions, that summarises the high-level functional properties of genes and transcripts. For example, biotype classes may annotate whether a gene is protein-coding or encodes a long non-coding RNA; if a gene is a disrupted homologue of a known protein coding gene (pseudogene) and by what mechanism it is produced; or the role of a small non-coding RNA such as post-transcriptional modification of other RNAs in the cell nucleus. We use biotype annotations to qualitatively visualize the clustering of gene embeddings into functional groups. We construct a multi-classification task using the top 10 most frequent biotype annotations as multi-class target labels which we predict from the unsupervised embeddings to assess how well biological function is encoded in the embedding space.

**Model & Training**   We use a frozen pretrained HyenaDNA model consisting of 8 layers and width 256 pretrained on sequences of length 160k. To extract sequence-level embeddings, we average along the sequence dimension in the final encoder layer. For comparison we also construct embeddings using DNABERT (5-mer) and Nucleotide Transformer. We construct embeddings for genes in the Ensembl dataset up to a length of 160k. For genes with sequence lengths exceeding the context

window of the encoder, we chunk the sequence and average the embeddings over the chunks. We utilize an XGBoost [7] classifier to perform the supervised multi-classification task on the embeddings. The hyperparameters used are shown in Table A.9.

Table A.9: **Hyperparameters.** Overview of XGBoost hyperparameters used in biotype multi-classifier.

| | |
|---|---|
| Estimators | 1000 |
| Max depth | 3 |
| Learning rate | 0.1 |
| Objective | softmax |

**Results**    As shown in 4.4, HyenaDNA achieves the highest F1 score on the biotype classification task indicating that its embeddings contain features that are informative of biological function. Notably, HyenaDNA achieves this using the much smaller embedding space dimension of 256, compared to DNABERT and Nucleotide Transformer, which produce embeddings of dimension 1029 and 1280, respectively.

## A.6    Long-range Species Classification Details

Table A.10: Hyperparameter ranges for ultra-long range species classification task. Transformer uses FlashAttention [11].

| | TRANSFORMER | | HyenaDNA | | | |
|---|---|---|---|---|---|---|
| Layers | 2 | 2 | 2 | 2 | 8 | 8 |
| Sequence length | 1024 | 32768 | 1024 | 32768 | 250000 | 450000 |
| Width | 128 | 128 | 128 | 128 | 256 | 256 |
| Parameters (M) | 0.5 | 4.5 | 0.4 | 0.4 | 6.6 | 6.6 |
| Num heads | 8 | 8 | - | - | - | - |
| Learning rate | $6e^{-5}$ | $6e^{-4}$ | $6e^{-5}$ | $3e^{-4}$ | $6e^{-5}$ | $6e^{-4}$ |
| Optimizer | AdamW | | | | | |
| Optimizer momentum | $\beta_1, \beta_2 = 0.9, 0.999$ | | | | | |
| LR scheduler | Cosine decay | | | | | |
| Weight decay (model) | 0.1 | | | | | |
| Weight decay (Hyena layers) | 0 | | | | | |
| Embed dropout | 0.1 | | | | | |
| Resid dropout | 0 | | | | | |
| Batch size | 128 - 256 | | | | | |
| Training epoch | 200 | | | | | |
| Reverse complement aug. | False | | | | | |

**Background**    Given a genetic sequence randomly sampled from a set of different species, successful identification of the source species requires a model to learn a distinct mutational profile for each species. The more locations for discriminative mutations a model can consider, the more successful it should be at this task. We can arbitrarily tune this task's difficulty by including a higher number of species or increasing the evolutionary similarity of the included species, and thus it represents a helpful setting for measuring long context reasoning abilities for DNA sequence models.

**Data**    We select five species for this task: human (*homo sapien*), lemur (*lemur catta*), mouse (*mus musculus*), pig (*sus scrofa*), and hippo (*hippopotamus amphibius*). We hold out four chromosomes from each species (chromosome numbers 1, 3, 12, and 13) for evaluation, and use the rest of each species' chromosomes for training.

**Model**    We compare HyenaDNA against a baseline Transformer, which uses Flash Attention [11] in the mixing layer instead of a Hyena operator. We use 2 and 8 layer models, depending on sequence length. For HyenaDNA, we train on sequence lengths of 1k, 32k, 250k, 450k and 1M. For

Transformer, we limit sequence lengths to 1k and 32k due to the quadratic increase in training time, making training infeasible on our hardware. See Table A.10 for model sizes and hyperparamters.

**Training**   We use pretrained models from 4.1, trained on various lengths between 1k to 1M nucleotides, and fine-tune them using a linear decoder head. We either pool across all tokens (1k and 32k models) or use the last token for classification (250k - 1M models). We randomly sample a *(species, chromosome, sequence start, sequence end)* tuple at each training step, with uniform probability across all species and non-held-out chromosomes. If a sequence's starting location on a chromosome is such that the end of that sequence would exceed the length of the chromosome, then we pad the sequence with N's to its full intended length. For evaluation, we randomly sample a *(species, chromosome, sequence start, sequence end)* tuple from our held-out evaluation set of chromosomes, and record the overall Top-1 5-way accuracy of our model (i.e. fraction of sequences correctly classified).

At sequence length 450k, we use the sequence length warm-up scheduler described in 3.2 on HyenaDNA. This involves gradually increasing the length of sequences fed to the model during fine-tuning from 1k to 450k. We observe better convergence and higher overall peak accuracy with this strategy, as shown in 3.2.

Table A.11: **Pretraining vs scratch on 5-way species classification.**   Top 1% accuracy for HyenaDNA by sequence length.

|  | HyenaDNA | |
|---|---|---|
| LENGTH | SCRATCH | PRETRAINED |
| 1k | 53.9 | 61.1 |
| 32k | 70.7 | 93.4 |
| 250k | 65.7 | 97.9 |
| 450k | 71.4 | 99.4 |

**Pretraining ablation**   For species classification, pretraining becomes more important for longer sequences. This is in-line with our observation that for harder tasks (including longer sequences), pretraining becomes more important. At sequence length 250k and 450k, the scratch vs. pretraining gap is 30+ accuracy points.

