# OpenReview forum: "HyenaDNA: Long-Range Genomic Sequence Modeling at Single Nucleotide Resolution"
_NeurIPS.cc/2023/Conference — NeurIPS 2023 spotlight_

### Official Review · Reviewer_JfVK · 2023-06-17

**Soundness:** 4 excellent
**Presentation:** 3 good
**Contribution:** 3 good
**Rating:** 7
**Confidence:** 3

**Summary:**

In this work, the authors propose a new genomic foundational model that can be pretrained on the human reference genome. This model, called HyenaDNA, is built upon the previous model Hyena. The advantage of this model is that it can train on ultra-long sequences (up to 450k) at single nucleotide resolution with significantly fewer parameters. The authors introduce a warm-up technique to stabilize the training procedure, and they also employ a soft prompting method for downstream tasks. The authors show the efficacy of their framework through various experimental settings.

**Strengths:**

1. This work tries to solve an important problem: developing new foundational models for DNA sequences.
2. Previous genomic foundational models are usually only able to be trained on relatively short sequences (<10kb). It is hard for them to deal with ultra-long sequences, which is necessary to capture long-range dependencies. This work provides a strategy to train the model on these long sequences.
3. The experimental results shown in the manuscript seem promising.
4. Overall, the manuscript is well written.

**Weaknesses:**

1. It is a bit difficult to understand 3.1 especially for people who are not familiar with the previous Hyena manuscript. $x_1$, $x_2$, $v$, $W_{x_{1}}$, $W_{x_{2}}$, $W_{v}$ are not defined in the main text. Line 159, not sure “input time” refers to. Line 162, not sure “projections” refer to.

Some minor points:
1. Section 2.1, no definition of D, is it equal to 4 representing {A,T,C,G}?
2. Line 320, A.5 should be A.4
3. Line 350, A.4 should be A.6
4. Appendix line 613, Fig. 4.2 should be Fig 4.1.
5. Line 206 equation, equation number missing

**Questions:**

1. Line 206, it is not so clear what the embedding step refers to. Also, perhaps to double check the correctness of $x\in\mathbb{R}^{L\times(T+N)}$
2. It seems that for different experiments, the authors train different HyenaDNA (different hyper-parameters, or using input sequences with different lengths as the pretraining data). Is there any specific reason that the authors use different models instead of e.g. using the biggest model that can handle up to 450k context, for all the experiments?
3. Table A.3, for different tasks the authors use different hyper-parameters, how these hyper-parameters be chosen?
4. The authors indicate in the main text their framework is more computationally efficient than Nucleotide Transformer since it has a significantly smaller number of parameters. However, in the appendix the authors mention that for each new task the entire model needs to be fine-tuned. How efficient this fine-tuning is compared to the fine-tuning procedure used in Nucleotide Transformer? How many computer resources are required for fine-tuning HyenaDNA?
5. The authors compare HyenaDNA with Nucleotide Transformer in 17 datasets. However, in the paper of Nucleotide Transformer there are 18 datasets in total. Why one dataset missing?

**Limitations:**

The authors have pointed out some limitations of their work in the conclusion section. I don’t think there is any significant negative societal impact of this work.

---

> ### Author Rebuttal · Authors · 2023-08-10
>
> We thank the reviewer for their thoughtful review! We’re glad you appreciate the importance of foundation models for DNA sequences, and the clarity of writing of the manuscript. Below we address concerns and clarifications the reviewer raised. We’re happy to answer any further questions the reviewer may have.
>
> ### Hyena method clarification
>
> Thank you for pointing this out. We’ll be sure to modify the description for readers unfamiliar with the original Hyena work as well. Specifically:
>
> Starting with the input $x$ (a length $L$ sequence of embeddings of size $D$), we apply three different parametric linear operators to obtain “projections” $x_1, x_2, v$, each of the same dimensions of the input. For example, $x_1 = T_{x1} x W_{x1}$ where $T_{x1}$ is the L x L Toeplitz matrix corresponding to a short convolution, and $W_{x1}$ a D x D weight matrix.
> After computing the projections, we start applying the Hyena operator recurrently, starting with $v$. First, we compute $ z = D_{x1} v$ where $D_{x1} = diag(x_1)$ (gating), following up with the long convolution $T_h$ and another gate $D_{x2}$. Higher-order Hyena operators (with more projections) continue until all projections have been used in the recurrence, following the same pattern of diagonal and Toeplitz.
> The phrase “input time” (on line 159) is meant to describe using time (or position in the sequence) as an input to the neural network parametrizing the convolution filter $h$. The inputs are the positions of the filter (kernel), and the outputs are the convolution weights for that position. Note:  the use of “time” is a remnant of signal processing literature (which Hyena draws from), but can also refer to space or a position.
> The term “projections” refers to a linear projection (applying a linear layer) to an input.
>
> ### Minor Points
>
> Thank you for catching the fine grained details that were missing here.
>
> 1. In section 2.1, D refers to the model embedding dimension. We’ll define this explicitly in the paper.
> 2. through 5., we will edit these typos and omissions in the final paper.
>
> ### Questions
>
> 1. The embedding step uses a standard embedding function (e.g. nn.Embedding in Pytorch). In general, this maps an index value (like 0 through 3, representing indexes of DNA nucleotides) to a learned embedding space of dimension D (typically the model dimension).
>
> 1a. Indeed, we do finetune different pretrained models of different sizes depending on the downstream task. There are 2 reasons for this:
>
> **a. Efficiency:**  Using the “right” size model that is sufficient for the downstream task is far more compute efficient, since ultralong-range sequence models require more training time, data and parameters to perform well.
>
> **b. Reduce overfitting:**  the larger models tend to overfit quicker (a common occurrence across machine learning). This is especially true if we’re testing on a set of short-range tasks (<1k long). Using a large model (eg for 450k sequences) will overfit quicker on these short range sequences if there is not a sufficient number of samples. Many of the short range genomic datasets are indeed small, typically 10s of thousands of samples, and as few as 1200 samples.
>
> 2. In Table A.3. For the Nucleotide Transformer datasets, we performed an extensive hyperparameter sweep via a grid search, a common hyperparameter search strategy. In practice, as we train, we use our intuition about which hyperparameters to adjust based on the training loss curves, for example, if regularization needs to be adjusted to reduce overfitting.
>
> 3. Regarding computation efficiency compared to baseline models, the Nucleotide Transformer authors also finetuned on every new downstream task. In comparison though, the Nucleotide Transformer used 8-A100-80GB GPUs for about 50 mins on average to finetune each dataset, as described in their paper. We used a single A100-40GB GPU, for 10 to 30 mins across the same datasets. Concretely, we provide results in Table R4 of the common response for a comparison of GPU-hours for pretraining and finetuning between baseline models and HyenaDNA. We will add these results to the appendix.
>
> 4. The 18th dataset of the Nucleotide Transformer (on splice site prediction) had a broken link during the time we were working on the submission. The dataset is now available publicly, and we have updated our results with this dataset (as shown in Table R2 of the common response).  Thanks for pointing this out. We slightly underperformed the Nucleotide Transformer on this task (by 0.4 accuracy points)
>
> Thank you again for the thorough review. Please let us know if there are other things we can clarify.

---

> > ### Comment · Reviewer_JfVK · 2023-08-13
> > **Thank you for your response**
> >
> > I would like to thank the authors for their response. I have one follow-up question regarding the computational efficiency of fine-tuning. The authors mention in their comments that "the Nucleotide Transformer uses a parameter-efficient finetuning: only 0.5% parameters are updated.". However, the original paper says that they only need 0.1% parameters for fine-tuning. And if I understand correctly, fine-tuning HyenaDNA requires all the parameters. Therefore I am a little bit surprised that in table R5, HyenaDNA can be much faster than NT-2.5B. In addition, the number of fine-tuned parameters of NT-2.5B should be significantly smaller than the number of parameters of DNABERT. However, as shown in Table R5, DNABERT is still significantly faster than NT-2.5B. Can the authors do some more elaborations on that?

---

> > > ### Author Response · Authors · 2023-08-14
> > >
> > > Thank you for the comments, and for catching the mistake. It is indeed 0.1% parameters that are updated for the Nucleotide Transformer (NT) model. We will update that number for the NT in Table R5 of the common response.
> > >
> > > Yes, it is also correct that HyenaDNA updates all its parameters during finetuning.
> > >
> > > **In regards to how HyenaDNA is much faster (in overall GPU-hours) than the NT-2.5B even though it only updates 0.1% of its parameters (requiring forward and backward passes):**
> > >
> > > - the NT model still needs to compute a very large forward pass on all its parameters, which consequently is still quite time consuming. As a rough benchmark, Li et al (2020) showed that a neural network (including BERT, like the NT model) can have a backward pass that is 3x longer than a forward pass (in latency). So although the NT model saves a lot of compute time by freezing most of its parameters (saving backward pass), its sheer size still requires a lot of compute (forward pass) compared to HyenaDNA (as well as DNABERT).
> > >
> > > - Another reason for the large difference between the NT model in Table R5 is that HyenaDNA is pretrained on far less data while achieving competitive (or better) results than the NT model. We provided Table R5 to compare total GPU-hours (at the request of a reviewer) to provide a different perspective on efficiency, which can be viewed in multiple dimensions.
> > >
> > > - Another efficiency dimension, in the original manuscript (Figure 4.1), is to control for the same model and data size, and compare the runtime for a small, 2 layer model with attention vs HyenaDNA. In that case, the main efficiency benefit of HyenaDNA is primarily at longer sequences, which grow near linearly ($N log N$) vs $N^2$ with attention.
> > >
> > > Thanks very much for the engaging and thoughtful questions.
> > >
> > > ### Citation
> > >
> > > Li, Shen, et al. "Pytorch distributed: Experiences on accelerating data parallel training." arXiv preprint arXiv:2006.15704 (2020).

---

> > > > ### Author Response · Authors · 2023-08-21
> > > >
> > > > We thank the reviewer for their time and the positive comments about the importance of the work! We wanted to share a friendly reminder that the discussion period is about to end tomorrow morning at 1PM EDT. We hope the reviewer is able to take into account our response, including additional ablation experiments and clarifications provided, into the final evaluation of the work. If there is no update, we certainly respect the reviewer’s decision. Thank you!

---

> > > > > ### Comment · Reviewer_JfVK · 2023-08-22
> > > > >
> > > > > I would like to thank the authors for their detailed responses. After reading the authors' as well as other reviewers' comments, I decide to increase my score by 1.

---

### Official Review · Reviewer_DWDg · 2023-07-06

**Soundness:** 3 good
**Presentation:** 3 good
**Contribution:** 3 good
**Rating:** 7
**Confidence:** 5

**Summary:**

This paper presents HyenaDNA, an advanced genomic foundation model leveraging the Hyena language model's capabilities, which are based on implicit convolutions. The authors highlight the limitations of prior Transformer-based genomic models, which have been constrained by token lengths and therefore impeded accurate modeling of long-range genetic interactions. These models also relied on tokenizers or fixed k-mers, which resulted in loss of single nucleotide resolution and crucial genetic variations. HyenaDNA addresses these issues. The model's unique abilities, including its sub-quadratic scaling in sequence length, usage of single nucleotide tokens, and full global context at each layer, mark a significant advancement in genomics.

**Strengths:**

Overall, this is an excellent paper! I appreciate the application of the Hyena model to DNA sequences. The utilization of advanced sequence modeling to tackle significant challenges in biology is commendable.

The concept of soft prompting for long-context models is exceptionally innovative and interesting!

**Weaknesses:**

- To be frank, I believe the evaluation of short sequences may not be crucial for this paper, considering its primary focus on long-range modeling. I would suggest including a comparison with the Enformer model for a more comprehensive perspective. More benchmarks on the long-context task  are necessary.

- Moreover, additional technical details should be incorporated either in the main text or the supplementary materials to facilitate a thorough understanding of the methodology used.

See more detail in the questions:

**Questions:**

I would raise my rating if the authors resolve my concerns.

- On line 132 , it would be beneficial to showcase $ T_{i, j} $.
- Could you elaborate a bit more on Figure 3.1, specifically, what the arrows represent?
- Could you provide detailed information about the baseline CNN model used in the GenomicsBenchmark?
- Are there any results from training the model from scratch? The connection between the pre-training goal and the downstream task doesn't seem very apparent.
- How well did the CNN model perform in predicting on the Nucleotide Transformer benchmark?
- On lines 259 - 260, how would training from scratch for each task fare? If the advantages of pre-training are minimal, using $3200 x$ less pretraining data doesn't present a significant benefit.
- Could you detail the fine-tuning process in section 4.2?
- For Figure 4.1, it would be helpful to incorporate the data from table 4.1 to display the "upper bound" performance.
- It would be advantageous to always include a training-from-scratch CNN or dilated CNN on all tasks.
- The species classification result is quite impressive! Would it be possible to apply any Explainable $\mathrm{Al}$ (XAI) methods to discern the principles that the model has learned?
- It would be beneficial to update the supplement to include more details about the Long Convolution. I had to refer to a previous paper to understand this concept.

---

> ### Author Rebuttal · Authors · 2023-08-10
>
>
> Thank you for the thoughtful review. We’re glad the reviewer is excited about the contribution of Hyena to DNA sequences, as well as the novelty of soft prompting for long-context. Below we address concerns the reviewer raised, and clarifications to technical details that were asked. We’re happy to answer any further questions the reviewer may have.
>
> ### Short vs long range evaluations
> The suggestion to apply HyenaDNA to the Enformer [Avsec et al., 2021] task is a good one. We dwelled on applying HyenaDNA to Enformer for quite some time. After much debate, we made the strategic choice to focus on a general foundation model (as opposed to a supervised Enformer task), with the compute budget we had. Our hope was that we could reach more computational biologists who needed more expressive long-range backbone models that could be easily adapted for their specific downstream tasks. We absolutely believe Enformer would be an incredibly exciting application of HyenaDNA and leave that to future research.
>
> That being said, we do showcase a number of long-range downstream tasks including chromatin profile (8k long), a novel ultralong-range species classification tasks up to 1M tokens, in-context learning tasks that uses over 32k tokens, and biotype classification on up to 160k tokens.
>
> We primarily included a high number of short-range benchmarks because of their accessibility and clear ability to compare against existing models. We hope that these make the case that HyenaDNA performs well on both short and long range tasks.
>
> ### Additional methodology details
>
> Thank you for bringing these up. We’ll be sure to add further architecture and training details to section 4 and appendix of the manuscript. We will also release all the code (with colab notebook examples), and model weights.
>
> ### Questions
>
> **Model and training clarifications**
>
> - Thanks for pointing these out. We will add clarifications regarding $T_{ij}$ on line 132, more details on the long convolutions in the supplement section, and add an upper bound to the tuneable tokens for table 4.1. $T_h$ is the Toeplitz matrix representation of a convolution with filter $h$. It is a mathematically equivalent representation ($T_h u$ or $h * u$), with definition $T_ij = h_(i - j)$ i.e. T is the matrix with the filter on each row, shifted by the column index.
>
> - We will amend section 3.1 for clarity (also see figure A.1 in the supplement for block diagram of HyenaDNA). To address your specific question, in Figure 3.1, the arrows represent how an input flows through a single Hyena operator (inside). Starting with the input $x$, separate projections are produced using 3 different $W$ matrices (dense layers). This is followed by 3 short convolutional layers, $T_{x2}$, $T_{x1}$, and $T_{v}$, creating $x_{2}$, $x_{1}$, and $v$.  Then $v$ is elem-wise gated by $D_{x1}$ (a diagonal matrix/layer), followed by a long convolution by $T_{h}$, and finally another gate by $x2$.
>
> - We will add further details in the appendix on the GenomicBenchmarks baseline CNN. As described by the original authors, the CNN uses an embedding layer, 3 conv layers with number of filters: 16, 8, and 4. It uses batch norm and max pooling after each conv layer, followed by 2 dense layers. It is trained for 10 epochs with batch size 64. The model sizes range from 120k to 520k, depending on sequence length chosen. We did not apply this basic CNN on the Nucleotide Transformer datasets as it appeared to be a simple demonstration model by the dataset authors. We’ll be glad to add these by the camera-ready if the reviewer believes this would bring clarity into how HyenaDNA performs over previous methods.
>
> - The finetuning procedure in section 4.2 has some additional details in the supplement information section (A.2), but we will certainly provide further details in the camera-ready. Specifically, our finetuning procedure attaches a single linear layer to a pretrained HyenaDNA model (2 layer, d_model=128 and 256), pretrained on sequence length 1k. The output embeddings learned are averaged over all the tokens, and then used to classify a class prediction. Each dataset is finetuned separately and sweeped through a number of hyperparameters (via a grid search). We provide a range of the hyperparameters in Supplemental Tables A.2 GenomicBenchmarks and Table A.3 for the Nucleotide Transformer datasets.
>
> **Scratch/pretraining**
>
> - Thank you for the suggestion - we provided additional experiments training from scratch in the common response in Tables R1 to R3, along with our observations. We will add these to section 4 of the manuscript. The key takeaways were that on simpler tasks, pretrained boosted mildly to moderately for HyenaDNA.
> - As difficulty increased (both the task itself and the sequence length), we observed greater performance gains from pretraining. See the common response and Table R1 and Table R3 for more in depth analysis of pretraining effects on downstream performance.
>
> **Species classification**
> We’re glad the reviewer found the species results impressive. We certainly think XAI methods can be applied to this task. Though our initial experiment focused primarily on showcasing the relative performance between longer sequence lengths, we think XAI methods applied to HyenaDNA would be an exciting future research direction, especially given the successful history of using convolutions in interpretability in genomics.
>
>
> Thanks so much for the thoughtful and very detailed feedback and questions, it’s helped us better communicate our work!
>
> ### Citations
>
> Avsec, Žiga, et al. "Effective gene expression prediction from sequence by integrating long-range interactions." Nature methods 18.10 (2021): 1196-1203.

---

> > ### Comment · Reviewer_DWDg · 2023-08-13
> > **concern about the short sequence benchmark**
> >
> > - The key concern is that I don't see much reason why the short sequence prediction is also improved, which is probably just because the CNN baseline is not so good (also, reviewer nDFb cannot reproduce this result). And based on the DeepSEA benchmark, I would say the difference is relatively small. The motivation should be the long context task from the design motivation view.
> > - For training from scratch, do you use the same hyperparameters to train or fine-tune the hyperparameters for training from scratch, such as the possibility that training from scratch may need longer training, a smaller learning rate? Also in the cross-specials benchmark, do you also use the warm-up?
> > - Could you please provide a time comparison between training from scratch and the pretrain+fine-tune setting? This could be a great comparison to know if pre-training is necessary.

---

> > > ### Author Response · Authors · 2023-08-14
> > >
> > > Thanks for your engagement, we appreciate your feedback and interesting questions.
> > >
> > > **1. Short range performance:**  We agree with the reviewer, the focus should be on the long range design and capabilities of HyenaDNA. The short range benchmarks are primarily meant as a comparison for common benchmarks, and to show HyenaDNA is at least as good as existing models. They are also readily available, and use low compute resources to enable most in academia to test it out.
> > >
> > > That being said, the short range models do allow us to test out our other design choices on DNA sequences. From our ablations in the common response Table R2, it appears other design choices also improve the short range tasks, including the single nucleotide tokenization, which is useful for any range.
> > >
> > > We also agree that the CNN is a weaker baseline. Another reviewer did suggest using a stronger baseline on the GenomicBenchmarks with DNABERT, which is now provided in Table R1, and which achieves SotA on 1 of 8 GenomicBenchmarks datasets.
> > >
> > > The Nucleotide Transformer is arguably a much stronger baseline model (previous SotA) and evaluated on harder tasks across 18 datasets, for which HyenaDNA performs competitively as well. For the DeepSEA benchmark, HyenaDNA’s main benefit is the much smaller model size while performing similarly. Notably, this dataset is nearly saturated, and so reducing the error rate becomes exponentially more difficult compared, for example, the NT datasets. We look forward to continue applying HyenaDNA to longer range benchmarks.
> > >
> > > **2. Training from scratch:**  For training from scratch, we sweep hyperparameters as well, though sometimes the best hyperparameters are the same for both scratch and finetuning. It did not appear that the best learning rates were significantly changed because of the finetuning vs scratch. What we did notice was that the training becomes much more stable with the pretrained models, i.e. the loss curves are smoother and less “jumpy”. This was especially true for the long range species classification task, where the ultralong sequences cause severe instability.
> > >
> > > **2.a Species warmup:**  Indeed, we use sequence length warmup on species classification for the ultralong-range (250k+).
> > >
> > > **2.b Reproducing results:** Please see the common response **[dated Aug 13]** to reviewer **[nDfb]**, in which we provide a Dockerfile that contains the exact settings and launch commands to reproduce 5 sample datasets from the Nucleotide Transformer. It appears the reviewer was missing the correct code, hyperparameters, and pretrained weights. We're confident the correct code (found in the Docker image) will reproduce the reported scores.
> > >
> > > **3. Pretrain vs scratch time:** Thank you for the suggestion. We have now added this information in Table R6 of the common response - comparing convergence times between scratch and pretrained finetuning on the Nucleotide Transformer datasets. Though they have similar training times for their respective top metrics (23 vs 26 mins on average per dataset), the pretrained models generally reach the same performance metric (as the scratch models) faster, and then continue to eke out more gains, albeit more slowly. We appreciate the suggestions and will put these results in the appendix of the manuscript.
> > >
> > > Thanks very much for the great follow-up questions.

---

> > > > ### Comment · Reviewer_DWDg · 2023-08-17
> > > > **Reply**
> > > >
> > > > Thanks a lot for the response! But considering the lack of other long sequence benchmarks, I would not change the score (especially since it's already weak Accept)

---

> > > > > ### Author Response · Authors · 2023-08-17
> > > > >
> > > > > We respect the reviewer's decision. We thank the reviewer for their time and the engaging discussion. We appreciate the suggestions on clarifying the methods, suggesting stronger baselines on GenomicBenchmark, and the suggested ablations on pretraining vs. scratch, and autoregressive vs bidirectional design choices. They've helped us identify and quantify how these designs led to improved results over previous methods.

---

> > > > > > ### Comment · Reviewer_DWDg · 2023-08-20
> > > > > > **Update the score**
> > > > > >
> > > > > > After considering more results from the training from scratch, it has resolved most of my concerns now, except for the lack of a long-sequence benchmark. So, as promised in my review, I am raising my score to 7.

---

> > > > > > > ### Author Response · Authors · 2023-08-20
> > > > > > >
> > > > > > > Thank you! We definitely appreciate the reviewer updating their score.

---

### Official Review · Reviewer_nDFb · 2023-07-06

**Soundness:** 3 good
**Presentation:** 3 good
**Contribution:** 3 good
**Rating:** 7
**Confidence:** 5

**Summary:**

This paper introduces HyenaDNA, a genome foundation model based on the Hyena architecture that replaces attention layers with implicit convolutions.  Though being 2500 times smaller, it achieves better performance than the state-of-the-art model.

**Strengths:**

- The paper is clearly written.
- The proposed method achieves very strong performance with very few parameters and is able to process ultra-long dna sequences.
- This paper investigates in-context learning and tunable prompting to the area of dna language model and show great performance improvements.

**Weaknesses:**

- The comparison over baselines is unfair.
    - Since Hyena is fundamentally different from the Transformer-based architecture, simply comparing the number of parameters does not accurate reflects the model efficiency. I think it would be more accurate the compare the wall-clock time and GPU time used in both pre-training and downstream evaluation. Actually, according to the figure A.2, the model is slower than its attention-based variants when inputs are shorted than $10^4$, which is very common in genome analysis tasks. Therefore, it is very interested to see how efficient the model is on each downstream task compared to DNABERT and Nucleotide Transformers.
    - All the previous models use standard or parameter-efficient fine-tuning for downstream evaluation, I think it is important to also fine-tune HyenaDNA on each downstream tasks to show how good it is as a *backbone* model and for fair comparison with other models. Also, it is interesting to see how the baseline models performs with the  tunable prompting.

- This paper introduces a very strong model with a series of fancy techniques, however, as an academical publication, it fails to provide enough insights and explanations to the community about the superiority of its performance.
    - According to the Hyena paper, the Hyena architecture achieves a similar level of performance (slightly better) as the transformer-based model (e.g., GPT) with the same number of parameters in general language modeling. Therefore, the fact that it dominants transformer-based models in the dna language modeling indicates the existence of **fundamental errors** of all the existing works. If this is true, what are the errors? I can think of a few possibilities:
        - Masked language modeling vs casual language modeling
        - k-mer tokenization vs character-level tokenization
        - attention mechanism fails to capture the semantics of DNA. (however, the AttnDNA performs very well)
    - To be honest, I don't think any of the above can lead to such significant improvements. Thus, throughout ablation study and experiments under the same setting is necessary.

**Questions:**

- How does AttnDNA perform on the datasets used by Nucleotide Transformers?
- Can you implement Nucleotide Transformer and DNABERT on the GenomicBank datasets for comparison?

**Limitations:**

In sum, the work is technically sound and empirically strong. However, based on the above thoughts, I tend to reject it for now. If the concerns are clearly solved, I would be happy to recommend a strong acceptance.

---

> ### Author Rebuttal · Authors · 2023-08-10
>
> We thank the reviewer for their detailed review of our work. We’re glad they appreciate the writing clarity, strong empirical results and the exploration of in-context learning and tunable prompting in genomics. Below we address the weaknesses and questions the reviewer described. We’re happy to follow up with any further clarifications the reviewer may have.
>
> ### Pretrain and Finetune Runtime Comparison
>
> Thank you for suggesting to compare runtime during pretraining and finetuning. We’ve provided these results for HyenaDNA and baseline models in Table R4 and R5 in the common response, and will add a section on efficiency in the appendix of the manuscript. We share part of the efficiency discussion we plan to include below as well.
>
> - Indeed, FlashAttention is faster than HyenaDNA on shorter sequences. Fortunately, HyenaDNA required far less pretraining time and data to more than make up for this, as noted by the high GPU-hour difference in Table R4.
>
> - We agree that many existing genomic tasks are shorter range, though we believe this to be due to the *constraints* of current sequence models - and that demand for long-range models is prevalent (eg gene expression, histone modification, splice prediction, chromatin accessibility and structure). As computational biologists see that longer-range models are possible, the emphasis on genomic benchmarks will likely shift to long-range tasks, shifting the advantage quickly to subquadratic scaling models.
>
> ### Insights into Hyena’s performance
>
> We’re glad the reviewer is interested in understanding where the HyenaDNA gains come from. We performed a series of ablations (details and results in Tables R1 to R3 of the common response) to help understand the effects of some of the factors the reviewer suggested, including:  k-mer tokenization vs single character, attention vs Hyena, and causal vs. bidirectional.
>
> The results of the ablation suggest each of these design choices contributed to gains over previous genomic FMs, and will be included in the manuscript in section 4 and the appendix. It may suggest fundamental errors (as the reviewer noted), or simply design choices that have yet to be explored in genomics and biology. We’re excited to further investigate other existing assumptions that may exist at the intersection of biology and machine learning.
>
> ### Finetune comparison (as a backbone model)
>
> We wanted to clarify that in all downstream tasks (except for section 4.3 on in-context learning), we perform standard finetuning with HyenaDNA. This enables a fairer comparison between previous genomic FMs. We hope this alleviates the reviewer’s concern about whether HyenaDNA is used as a backbone in a similar fashion to baselines.
>
> **Tuneable prompting on Baseline models:**  For comparing baselines models with tuneable prompting, this unfortunately would require a significant redesign of their models. The most important constraint is the context length, which HyenaDNA opens up, but previous methods are limited to 512 or 1000 tokens. HyenaDNA uses up to 32k tuneable tokens, in addition to adding multiple samples within the context, which would simply not be possible on DNABERT and Nucleotide Transformer. That may be a really interesting future research direction, and we hope we’ve inspired others to pursue this.
>
> ### Questions
>
> **1. AttnDNA on the Nucleotide Transformer (NT) datasets:** As suggested by the reviewer, we added results for AttnDNA on the NT datasets in the common response in Table R2. In all tasks, AttnDNA underperforms HyenaDNA (see common response for more analysis).
>
> To summarize the main takeaways, AttnDNA was able to perform well on simpler tasks ( promoter prediction), but struggled significantly on the more challenging histone mark tasks (the ones starting with “H”) as compared to HyenaDNA, and with very large gaps. This suggests that the Hyena operator itself also contributes to the boost in performance over previous attention-based genomic FMs.
>
> **2. DNABERT finetuning:** As suggested, we finetuned DNABERT as a stronger baseline on the GenomicBenchmarks, shown in the common response in Table R1. Indeed DNABERT does reach SotA on 1 of 8 datasets, while HyenaDNA retains top performance on 7 of 8 datasets.
>
> **Planned:** We plan to finish finetuning the Nucleotide Transformer model on the GenomicBenchmarks in the coming weeks. As these models are very large (500M to 2.5B parameters), we did not yet have the compute resources to complete training. We did finetune 1 of the 8 datasets for the Mouse Enhancer task. The 500M Nucleotide Transformer underperformed HyenaDNA by about 10 accuracy points (85.1 vs 75.2). If accepted, we can include the remaining 7 datasets in the camera-ready.
>
>
> Thank you again for the suggestions to improve our work, they’ve helped us better understand and communicate the differences between our methods and previous genomic FMs.

---

> > ### Author Response · Authors · 2023-08-10
> >
> > ***See comments 3/3 of common response for the analysis of AttnDNA finetuning on the Nucleotide Transformer datasets.***

---

### Official Review · Reviewer_LTPa · 2023-07-06

**Soundness:** 4 excellent
**Presentation:** 3 good
**Contribution:** 4 excellent
**Rating:** 8
**Confidence:** 4

**Summary:**

The authors train the Hyena operator model on the human genome and adapt it to downstream tasks in computational biology.


**Strengths:**

Lots of clever things about this work that I really like:

Very good use case of Hyena model with very long-range dependencies
Curriculum learning is very clever and makes sense. Figure 3.2 is great to show this
I think Table 4.4 with the use of XGBoost is very simple but effective in getting a statistic about the utility of an embedding. With XGBoost, you are going to get a pretty reasonable and comparable every time you use it.

The model capacity shown by perplexity and sequence length is good.

I really really like that it is able to work at single base pair resolution.

I’m impressed and happy that section 4.4.3 uses held-out chromosomes for training and testing (rather than random sampling). Since you are working with multiple species, are there still some aspect of data leakage in train/test sets because of synteny?

I think this is impressive being only trained on the human genome. I’m excited to see this model trained on all genomes!


**Weaknesses:**

I think the GenomicsBenchmark claims about SotA could be improved. Definitions of SotA should be based on architecture, not the datasets. Since the dataset is newer than previously benchmarked models, I would recommend the authors evaluate the GenomicsBenchmark across other SotA models or architectures (like DNABERT and Nucleotide Transformer, which are mentioned in this paper). Alternatively the SotA claim is weak at best, and misleading at worst.

I think section 4.4.3 could have a very simple baseline of a kmer count model. Can a simple kmer based model do just as well, or poorly? I’m sure it is very fast.


**Questions:**

Figure 3.2 - Do longer sequences simply take longer to do a forward and backward pass? You show wall time. Are the number of steps relatively closer?

How much is the result of your model’s result is its architecture versus using tunable prompting? What would happen if you used tunable prompting for the Nucleotide Transformer?

How are the F1 and MCC scores for Table 4.2 generated? Which values are you comparing?

How was the finetuning in section 4.4.1 done? Is it not soft-prompting?


**Limitations:**

Generally, I find the term “Foundation Model” to be ambiguous and unnecessarily buzzwordy. HyenaDNA is a large, pretrained model. Even though the authors have pretrained the model, it has to be finetuned and quite heavily engineered differently for each subtask. But, I guess this is what the field is wanting to say.

While I do like that the important aspects of the paper are emphasized, some of the color, boldness, different font, and italicization can be toned down a bit. For example: “Therefore, having both long-range context and single nucleotide resolution simultaneously” doesn’t need to be blue and bolded. Also, what is the author’s intent in using blue text versus bold text?

---

> ### Author Rebuttal · Authors · 2023-08-10
>
> We thank the reviewer for their detailed review of our work and are glad they appreciate many of the contributions including the long-range capabilities, single nucleotide resolution, and the curriculum learning introduced. Below, we address the concerns the reviewer made about the GenomicBenchmarks baselines, and include additional ablations requested.
>
> ### GenomicBenchmark Baselines
>
> In the common response above, we added ablation results to address concerns the reviewer (and others made), including:
>
> - Finetuning a stronger baseline using DNABERT, which achieved SotA on 1 of 8 datasets (with HyenaDNA outperforming on the other 7 datasets).
> - HyenaDNA trained with a K-mer (K=6) tokenizer to compare the effect of K-mers vs single character tokenizers. See Table R1 in the common response. The K-mer tokenizer consistently degrades performance over the single nucleotide tokenizer (2-10 accuracy points).
>
> **Planned:** We plan to finish finetuning the Nucleotide Transformer model on the GenomicBenchmarks in the coming weeks. As these models are very large (500M to 2.5B parameters), we did not yet have the compute resources to complete training. We did finetune 1 of the 8 datasets for the Mouse Enhancer task. The 500M Nucleotide Transformer underperformed HyenaDNA by about 10 accuracy points (85.1 vs 75.2). If accepted, we can include the remaining 7 datasets in the camera-ready.
>
> ### Questions
>
> **Longer sequence time:** Indeed, longer sequences do require more time for forward and backward passes.  See figure 4.1 in the manuscript for the wall time (forward and backward pass) by sequence length.  We compare times for both HyenaDNA and its attention counterpart.
>
> **Data leakage / synteny:** That’s a really interesting point. It’s possible “data” leakage from this evolutionary process can occur, and it would be challenging to account for explicitly. We take comfort in that in the aim of the species classification task, we’re interested in the relative performance between long sequence lengths, over the absolute performance (in this example).
>
> **Tuneable Prompting vs. Finetuning:** To be clear, all of our short-range tasks in section 4.2, as well as 4.4, use **standard finetuning**.  This includes the chromatin profile task in 4.4.1 (as inquired by the reviewer). This makes the comparison between previous architectures more clear.  The tuneable prompting approach is a standalone set of experiments in section 4.3 only, and considered “self-contained”. We’ll be sure to make this distinction more explicit in the manuscript, thank you for pointing this out.
>
> **Tuneable prompting on Nucleotide Transformer:**  For comparing baseline models with tuneable prompting, this unfortunately would require a significant redesign of their models. The most important constraint is the context length, which HyenaDNA opens up, but previous methods are limited to 512 or 1000 tokens. HyenaDNA uses up to 32k tuneable tokens, in addition to adding multiple samples within the context, which would simply not be possible on DNABERT and Nucleotide Transformer. That may be a really interesting future research direction, and we hope we’ve inspired others to pursue this.
>
>
>
>
> **Foundation models terminology:** We acknowledge that this can be a contentious terminology to some. We took the stance that most effectively conveys what we hope and believe our model can do, which is to learn useful and general representations that can be applied to downstream tasks.
>
> **Styling / bold font:** Thank you for bringing this up, we initially wanted to bold and highlight text we thought were key takeaways per section. We’ll be sure to readdress the format as to not be so distracting.
>
> **F1 and MCC metrics:** The 18 datasets in the Nucleotide Transformer were sequence-level classification tasks (binary and 3-way). Using the model prediction for a DNA sequence, we’re able to determine a true/false positive or true/false negative with the label. F1 and MCC metrics are standard statistical methods used by the original authors of the Nucleotide Transformer.
>
> Formally, these are calculated as:
>
> $$F1 = 2 \cdot \frac{precision \cdot recall}{precision + recall}$$
>
> $$ MCC = \frac{TP \cdot TN - FP \cdot FN } {\sqrt{(TP+FP)(TP+FN)(TN+FP)(TN+FN)}} $$
>
> Implementation-wise, we used predefined metrics from the scikit-learn package to calculate our F1 and MCC metrics. For the F1 score, we used the f1_score function [https://scikit-learn.org/stable/modules/model_evaluation.html#precision-recall-f-measure-metrics] with the average flag set to macro. And for the MCC values, we used the matthews_corrcoef function [https://scikit-learn.org/stable/modules/model_evaluation.html#matthews-corrcoef].

---

> > ### Comment · Reviewer_LTPa · 2023-08-15
> > **Follow up**
> >
> > Thank you for your comments and continued discussion.
> >
> > I will keep my score the same, so long as the claims about the GenomicsBenchmark in the abstract and text are updated.

---

> > > ### Author Response · Authors · 2023-08-15
> > >
> > > We appreciate your time!  We will update the GenomicBenchmarks claim as you have suggested. Thank you.

---

### Official Review · Reviewer_YJ4C · 2023-07-07

**Soundness:** 3 good
**Presentation:** 3 good
**Contribution:** 2 fair
**Rating:** 7
**Confidence:** 4

**Summary:**

This manuscript applies Hyena, a neural operator based on implicitly parametrized long convolutions and data-controlled gating, to the domain of DNA modelling.
The subquadratic complexity of Hyena enables scaling to context lengths of up to 450,000 at single nucleotide resolution.
This represents a significant improvement over previous methods based on dense attention, increasing pre-training sequence length by more than two orders of magnitude.
Pre-training is done using a sequence length warm-up scheme, such that sequences grow longer as training progresses.
The resulting model is benchmarked on a variety of downstream tasks using different adaptation methods.

**Strengths:**

1. (**Originality**) HyenaDNA is the first work to apply Hyena, a novel breed of implicitly parametrized long convolutions, to DNA modeling.
2. (**Originality**) The authors experiment with two adaptation strategies commonly used in NLP to apply the pre-trained model to novel tasks.
3. (**Quality**) The method is evaluated on a large number of genomic datasets introduced as part of related work. This firmly situates HyenaDNA within the existing literature.
4. (**Significance**) The proposed model outperforms state-of-the-art attention-based methods on the majority of presented downstream tasks with considerably fewer parameters and scales to extremely large context sizes of 450,000 tokens at single nucleotide resolution.
5. (**Clarity**) Overall, the manuscript is written clearly, and the arguments are easy to follow.
6. (**Clarity**) Architecture and hyperparameter details are clearly presented.
7. (**Clarity**) Including background task descriptions for the Nucleotide Transformer, the Biotype Classification, and the Chromatin Profile Prediction benchmarks are helpful for understanding the evaluation protocol.

**Weaknesses:**

1. (**Originality**) Beyond the application of Hyena to the DNA domain, the manuscript offers few new technical insights that could be of interest to the machine learning community. Sequence length warm-up is an established technique to stabilize training instabilities, and soft prompting, as well as few-shot fine-tuning, are commonly used adaptation strategies in the natural language processing community.
2. (**Quality**) Results are presented without any measure of deviation over multiple repetitions. This is true even for smaller scale fine-tuning experiments.
3. (**Quality**) For the GenomicsBenchmark, the authors should report more details about the CNN baseline (parameter count, number of layers, etc.), as well as training details and hyperparameters for AttnDNA. If the same hyperparameters are used for both HyenaDNA and AttnDNA, please state this explicitly.
4. (**Clarity**) Few-shot adaptation includes a tuning phase, making this adaptation method more similar to few-shot fine-tuning than in-context learning. Relevant literature should be cited.
5. (**Clarity**) The results reported in Table 4.1 for the Nucleotide Transformer do not seem to match with those reported in the original paper in Supplementary Table 6. Please explain this deviation.

**Questions:**

- Why is HyenaDNA pre-trained autoregressively rather than non-autoregressively, as was done in Nucleotide Transformers? Both methods would be feasible since downstream adaptation of HyenaDNA always includes a fine-tuning phase.
- Considering the limited vocabulary of just 4 nucleotides, it's somewhat unexpected that the reported perplexity score for a high-performing long-context model like HyenaDNA would be this high. Could you provide some insight on this? To underscore the significance of pre-training further, it would be valuable if you could supplement the evaluation of pre-trained embeddings using probing classifiers with an additional experiment where HyenaDNA is trained directly from scratch on the downstream datasets.
- The manuscript mentions that both a long context and single nucleotide resolution are crucial. However, K-mer tokenization could help reduce the temporal dimension, and previous work has shown that a higher k leads to improved results (Ji et al., 2021). Therefore, it would be beneficial to include an experiment that compares HyenaDNA models pre-trained with different tokenization schemes.

### References
- Ji, Y., Zhou, Z., Liu, H., & Davuluri, R. V. (2021). DNABERT: pre-trained Bidirectional Encoder Representations from Transformers model for DNA-language in genome. Bioinformatics, 37(15), 2112-2120.

### Minor comments

- Possible Typo in Table 4.1 row 2 (Coding vs. Intergenomic): the improvement over baseline performance (+3.5) does not match with the reported base performance. Both Base and HyenaDNA are reported to have the same accuracy.
- Typo on line 543 : sequence-level instead of sequence-leel
- Typo on line 544: that instead of taht

**Limitations:**

The authors address the limitations of their method and experiments in the manuscript.
Possible malicious uses of HyenaDNA are not addressed.

---

> ### Author Rebuttal · Authors · 2023-08-10
>
> We thank the reviewer for their detailed review of our work. We’re glad they appreciate the strength of the results, the originality and significance of the application, and the clarity of the manuscript. Below, we address the reviewer's concerns about the contributions, design, performance, and tokenization.
>
> ### ML community contributions
>
> This work is representative of many problems in ML, namely on how to efficiently capture statistics that include both local and long-scale dependencies, local spatial structure of objects and large-scale correlations of those objects in images, local motion and long action sequences in video, and structure within phrases and long-range correlation in text.
>
> Regarding technical relevance, we would like to highlight a few observations and insights we think are relevant to the ML community, and that could be used as a recipe for similar tasks.
>
> **Ultralong sequence training:** Training on 1M context length data gives rise to unique instability challenges, which depart from the typical scaling of model size or datasets. Sequence length warmup has been explored in the NLP community in the past, notably by only up to 2k long sequences [Li et al., 2022] and for small Transformers [Press et al., 2020]. Many questions remain about the schedule of increases (in particular for long sequences), learning rate adjustment, and how it affects token efficiency at ultralong sequences. We share one set of strategies and findings in genomics that were performant.
>
> **Single character training:** In nLP, the use of single characters consistently performs worse than aggregated tokens [Yu et al., 2023, Tay et a., 2021, Kalchbrenner et al., 2016]. Our work contributes a successful recipe for single character training that surpasses aggregated-character methods.
>
> **Model vs vocabulary size:** Our work raises novel questions on the tradeoff between model size vs vocabulary size. Given that our design uses a small vocabulary and model size (orders of magnitude smaller) than previous SotA genomic FMs, it begs the question of whether a similar design (single character tokens with Hyena) can be applied to natural language, and if these findings can be transferred.
>
> ### Clarity concerns
>
> - **GenomicBenchmarks CNN baseline details:**  We will include more details of the CNN baseline model in the appendix section A.2.1. As described by the original authors, the CNN uses an embedding layer, 3 conv layers with number of filters: 16, 8, and 4. It uses batch norm and max pooling after each conv layer, followed by 2 dense layers. It is trained for 10 epochs with batch size 64. The mode sizes range from 120k to 520k, depending on sequence length chosen.
>
> - **Hyperparameters:** We indeed use different hyperparameters between HyenaDNA and AttnDNA, and will clarify this in supplement section A.2.1.
>
> - **Few-shot adaptation vs in-context learning:**  We will make the distinction more clear with citations.
>
> - **Nucleotide Transformer Table 4.1 results:**  Thank you for pointing out the discrepancy. The Nucleotide Transformer paper has 2 sets of results: one in Figure 2 of the main paper, and one in Supplementary Table 6. We originally used Figure 2 in the main paper, but after speaking with the authors, the Supplementary Table 6 indeed does fit our procedure more closely by using the best test set results. We will update values in our comparison in the manuscript, and is already reflected in the common response in Table R2. Notably, this does not change the ranking of SotA performance per dataset for HyenaDNA.
>
> ### Questions
>
> **1. Autoregressive vs bidirectional:** HyenaDNA was pretrained autoregressively (causal) because the initial results showed better downstream performance over a bidirectional Hyena (see ablation in our common response) that we experimented with. The autoregressive training also offered additional appealing properties:
>
> - The Hyena model is naturally flexible to variable length sizes over BERT-style models that need to be explicitly fed variable lengths during training, since the causal model “sees” different lengths incrementally.
>
> - Additionally, we wanted to explore the use of in-context (ICL) learning in genomics, as we believe ICL has driven a lot of innovation in natural language. An autoregressive model is more amenable to current ICL methods that leverage next token prediction for class prediction, for example.
>
> **2. DNA perplexity**: We too were surprised by the relatively high perplexity for the given vocabulary size. We think this is a challenge inherent to the domain of genomics, and provide a few relevant data points:
>
> - Biologically, genomes carry “junk” DNA that is difficult to decipher and or not informative. One rationale for why this occurs can be viewed through an evolutionary lens, in that, as mutations occur over time, a mutation is less deleterious if it occurs in “junk” DNA, as opposed to informative DNA [Zhang el al., 2012, Ohno et al.,1972]. Knowing whether a sequence is “junk” (informative) or not is still an open question, making genomics overall particularly challenging.
>
> - [Rajarajeswari et al] showed DNA compression algorithms that sought to reduce bits per character (BPC, convertible to perplexity) in a similar range. They achieved an equivalent perplexity of 1.58 vs our 1.54, with a lower score as better.
>
> - [Benegas et al., 2022] trained a genomic plant Transformer (not FM) using single nucleotides, and observed similar perplexity.
>
> **3. Pretraining vs. scratch:**  Please see the common response for ablations on pretraining vs. finetuning.
> **4. K-mer tokenization:** In the ablation presented in the common response in Table R1, we observed consistent (and significant) degradation in performance on the GenomicBenchmarks when using 6-mer tokenization on HyenaDNA and training from scratch, from 2 to 10 accuracy points.
>
> **Limitations**
> We will address the potential for malicious uses of HyenaDNA in the camera-ready.

---

> > ### Author Response · Authors · 2023-08-10
> > **Continuation of rebuttal response**
> >
> > ### Continuation of question responses
> >
> > **4. K-mer tokenization:** We thank the reviewer for this suggestion. In the ablation presented in the common response in Table R1, we observed consistent (and significant) degradation in performance on the GenomicBenchmarks when using 6-mer tokenization on HyenaDNA and training from scratch, from 2 to 10 accuracy points.
> >
> > **Typos**
> >
> > We thank the reviewer noting the typos, and we will make the necessary changes.
> >
> > **Limitations**
> >
> > We will address the potential for malicious uses of HyenaDNA in the camera-ready. We believe it is an important conversation to have for any potentially powerful and widespread technology.
> >
> > ### Citations
> >
> > Li, Conglong, Minjia Zhang, and Yuxiong He. "The stability-efficiency dilemma: Investigating sequence length warmup for training GPT models." Advances in Neural Information Processing Systems 35 (2022): 26736-26750.
> >
> > Press, Ofir, Noah A. Smith, and Mike Lewis. "Shortformer: Better language modeling using shorter inputs." arXiv preprint arXiv:2012.15832 (2020).
> >
> > Benegas, G., S. S. Batra, and Y. S. Song. "DNA language models are powerful zero-shot predictors of non-coding variant effects." (2022).
> >
> > Rajarajeswari, Pothuraju, and Allam Apparao. "DNABIT compress–genome compression algorithm." Bioinformation 5.8 (2011): 350.
> >
> > Zhang, Zhe, et al. "Analyzing effects of naturally occurring missense mutations." Computational and mathematical methods in medicine 2012 (2012).
> >
> > Ohno, Susumu. "So much" junk" DNA in our genome. In" Evolution of Genetic Systems"." Brookhaven Symposium in Biology. Vol. 23. 1972.
> >
> > Tay, Yi, et al. "Charformer: Fast character transformers via gradient-based subword tokenization." arXiv preprint arXiv:2106.12672 (2021).
> >
> > Kalchbrenner, Nal, et al. "Neural machine translation in linear time." arXiv preprint arXiv:1610.10099 (2016).
> >
> > Yu, Lili, et al. "Megabyte: Predicting million-byte sequences with multiscale transformers." arXiv preprint arXiv:2305.07185 (2023).

---

> > > ### Author Response · Authors · 2023-08-21
> > >
> > > We thank the reviewer for their time and their comments! We wanted to share a friendly reminder that the discussion period is about to end tomorrow morning at 1PM EDT. We hope the reviewer is able to take into account our response, including additional ablation experiments and clarifications provided, into the final evaluation of the work. If there is no update, we certainly respect the reviewer’s decision. Thank you!

---

> > > ### Comment · Reviewer_YJ4C · 2023-08-22
> > >
> > > First, I'd like to thank the authors for their comprehensive response and the additional ablations.
> > >
> > > A context of 1 million tokens is indeed impressive and offers significant potential for genomic applications.
> > > However, as other reviewers have noted, the degree to which such an extended context is exploited needs to be explored further on long-range benchmarks.
> > > Given the efficiency of the proposed method, I am confident that we will see more in-depth analyses in the future.
> > >
> > > Thank you for addressing the discrepancies between your results and those reported in the Nucleotide Transformer paper.
> > > It remains surprising that a compact model like HyenaDNA can consistently outperform billion-parameter networks.
> > > This is counterintuitive to the general trends observed across the deep learning literature.
> > > Additionally, I appreciate the authors providing access to model weights and code, ensuring successful reproduction of the stated results.
> > > Exploring the inductive biases of Hyena or identifying the weaknesses of existing methods will definitely be interesting.
> > >
> > > Similar to the small size, using character-level tokenization contradicts findings from the natural language processing community and DNABERT.
> > > However, as your newly added results suggest, this appears to be a crucial design choice worth considering in genomic applications.
> > >
> > > Regarding DNA perplexity, I appreciate the detailed discussion and the added references.
> > > Reports of similar perplexities in the compression and genomic sequence modeling literature, combined with your experiments comparing pre-trained vs. from-scratch Hyenas, bolster my confidence that pre-training is essential.
> > > This reinforces the foundation model argument presented in the paper.
> > >
> > > Lastly, I'm grateful to the authors for addressing the dual-use nature of this work in the camera-ready version.
> > >
> > > Your thorough response and robust results on additional ablations have addressed many of my initial concerns.
> > > Therefore, I have decided to increase my original score from 6 to 7.

---

### Author Rebuttal · Authors · 2023-08-10

### Common Response

We thank the reviewers for their time and in-depth reviews. We believe that addressing the reviewer’s feedback and questions has helped greatly improve the quality of our manuscript.

We are happy to hear the reviewers appreciated the strong empirical performance of HyenaDNA over previous methods **[YJ4C, LTPa, nDFb]**, the novelty/relevance of the application **[YJ4C, LTPa, nDFb, DWDg, JfVK]**, and the clear description of the methods **[YJ4C, nDFb]**.

### Ablations

Multiple reviewers suggested ablations, including downstream performance for scratch vs. pretrained models **[YJ4C, DWDg]**, K-mer vs. single character tokenizers **[YJ4C, LTPa, nDFb]**, causal vs bidirectional **[YJ4C, nDFb]**, and additional model baselines on the GenomicBenchmarks and Nucleotide Transformer datasets **[LTPa, nDFb]**.

Below, we provide a common response with updates and ablation results that were requested by reviewers **[YJ4C, LTPa, nDFb, DWDg]**. In these experiments, we further investigate how each design choice in HyenaDNA contributes to performance gains compared to baseline models.

### Updates

Since our submission, we are pleased to report that we have pretrained an even longer HyenaDNA model: 1M context length (vs 450k in the submission), 500x longer than previous genomic FMs and is 160x faster than attention.

The results from ablations are in the PDF, with a summary below:

- Several new choices in HyenaDNA contribute to gains over previous genomic foundation models (FMs): single character vs. k-mer tokenizer, Hyena operator vs. attention, and the causal vs. bidirectional Hyena.
- Pretraining has a greater effect on the more challenging tasks, and as sequences become longer, eg, by up to 30 accuracy points on species classification at 450k context.
- HyenaDNA uses far less compute for pretraining and finetuning. E.g. the Nucleotide Transformer uses 128 A100 GPUs to pretrain for 1 month, while HyenaDNA used 1 A100 for 80 mins to pretrain a model used on the same downstream tasks.

**We provide further details on the ablations in the sections below.**

### Scratch / Pretraining

Reviewers **[YJ4C, DWDg]** inquired about how much pretraining (vs. scratch) improves downstream performance. To address this, we include training from scratch on 3 groups of datasets: GenomicBenchmarks, Nucleotide Transformer, and Species Classification.

**1. GenomicBenchmarks (Table R1 PDF):** Pretraining boosted HyenaDNA by up to 3.5 acc points, and by 1.8 points on average. HyenaDNA already performed strongly from scratch, which made pretraining gains more difficult. For AttnDNA, pretraining is more important, boosting performance by up to 11.7 acc points and by 4.5 points on average.

**2. Nucleotide Transformer datasets (Table R2 PDF):**
On the Nucleotide Transformer datasets: On the more challenging tasks (histone marks, datasets starting with “H”), pretraining boosts HyenaDNA metrics by up to 21 MCC points on H3K4me3. For simpler tasks (with higher baseline values) such as the splice sites and promoter tasks, there was less boost (0 to 1 accuracy points). Note: the 18 tasks use a mix of MCC, F1, and accuracy metrics, and so an average comparison is less meaningful.

**3. Long-range species classification (Table R3 PDF):**
For species classification, pretraining becomes more important for longer sequences, addressing questions from reviewers **[YJ4C and DWDg]** about where pretraining helps in HyenaDNA. At sequence length 250k & 450k, the scratch/pretrain gap is 30+ accuracy points.

### K-mer tokenization vs single nucleotides

To ablate the impact of the K-mer tokenizer vs. single character **[YJ4C, LTPa]**, we used the same K-mer (K=6) tokenizer from the Nucleotide Transformer model, which had a vocabulary of ~4100. We then trained a scratch HyenaDNA model using this K-mer tokenizer on the GenomicBenchmarks. The K-mer tokenizer degraded performance for every dataset, from 2 to 10 accuracy points compared to (scratch) HyenaDNA with a single character tokenizer (see Table R1 PDF). The K-mer tokenizer is one factor in HyenaDNA’s gain, which we will add to section 4.2.

### Bidirectional vs Causal

To ablate the impact of using a causal model **[YJ4C, nDFb]**, we implemented a bidirectional version of HyenaDNA and trained from scratch on the GenomicBenchmarks. The bidirectional version degraded performance on 7 of 8 datasets compared to the standard causal HyenaDNA (also from scratch), on average by 3.8 accuracy points. See Table R1 PDF. The bidirectional HyenaDNA was implemented by using a circular FFT convolution.

### DNABERT baseline

To compare against a stronger baseline on the GenomicBenchmarks suggested by reviewers **[LTPa and nDFb]**, we also finetune DNABERT. DNABERT is able to reach SotA on 1 dataset (Coding vs Intergenomic), and match on another (Human Enhancers Cohn) with HyenaDNA (Table R1 PDF). DNABERT uses 110M params, while HyenaDNA uses just 400k param. We will add these results to section 4.2.

### AttnDNA finetuning

We finetune the AttnDNA model on the Nucleotide Transformer datasets, Table R2 PDF. AttnDNA and HyenaDNA are causal and use single nucleotide tokens, but AttnDNA significantly underperformed against its Hyena counterpart. This suggests the Hyena operator itself contributes significantly to the overall performance gains of HyenaDNA.

### Pretrain and Finetune Runtime Comparisons

**Pretraining compute:** Reviewers **[LTPa, nDFb, JfVK]** suggested also comparing efficiency by compute resources used (in addition to parameter count). When comparing actual GPU-hrs used for pretraining across baseline models, HyenaDNA is more efficient than baselines. See Table R4 PDF. We will put a full table of results in the updated manuscript.

**Finetuning compute:** We use the GenomicBenchmarks for finetuning, and record the per epoch runtime in Table R5 PDF. The Nucleotide Transformer uses a parameter-efficient finetuning: only 0.5% parameters are updated.

---

> ### Author Response · Authors · 2023-08-10
> **Tables in markdown format (same as PDF)**
>
> ### Rebuttal 2/3
>
> ***We provide the tables in markdown format here for convenience as well.  These are the same as those in the PDF.***
>
> ***
>
> ### GenomicBenchmarks Ablations
>
> | Dataset | Baseline CNN (scratch) | AttnDNA (scratch) | AttnDNA (pretrain) | HyenaDNA (scratch) | HyenaDNA (pretrain) | HyenaDNA k-mer (scratch) | HyenaDNA Bidirectional (scratch) | DNABERT (pretrain) |
> | ----------- | ----------- | ----------- | ----------- | ----------- | ----------- | ----------- | ----------- | ----------- |
> | Mouse Enhancers  | 69  | 79.3  |  79.3  |  84.7  |  **85.1**  | 81.8  | 80.6 | 66.9 |
> | Coding vs Intergenomic   | 87.6  | 89.3  | 91.2 | 90.9 | 91.3 | 86.7 | 90.3 | **92.5** |
> | Human vs Worm  | 93.0  | 94.8 | **96.6** | 96.4 | **96.6** | 92.9| 95.9 | 96.5 |
> | Human Enhancer Cohn | 69.5  | 67.7 | 72.9 | 72.9 | **74.2**| 69.8 | 72.1 | 74.0 |
> | Human Enhancer Ensembl | 68.9  | 79.0 | 88.3 | 85.7| **89.2** | 88.0 | 85.9 | 85.7 |
> | Human Regulatory | 93.3  | 90.2 | 91.8| 90.4 | **93.8** | 90.2 | 89.1 | 88.1 |
> | Human Nontata Promoters | 84.6  | 85.2 | 90.1 | 93.3 | **96.6** | 83.5 | 88.5 | 85.6 |
> | Human OCR Ensembl | 68.9  | 68.3 | 79.9 | 78.8 | **80.9** | 70.2 | 75.3 | 75.1 |
>
> **Table R1. GenomicBenchmarks ablation results comparing HyenaDNA design components.**
>
> ***
>
> ### Nucleotide Transformer Ablation
>
> | Dataset | NT 2.5B | AttnDNA (pretrain) | HyenaDNA (pretrain) |  HyenaDNA (scratch) |
> |------|------|------|------|------|
> | Enhancer | 58.0 | 59.3 | **59.7** | 58.6 |
> | Enhancer types | 47.4 | 51.9 | **56.7** | 48.4 |
> | H3 | 81.4 | 75.8 | **82.3** | 79.9 |
> | H3K4me1 | 55.9 | 38.7 | **56.7** | 43.4 |
> | H3K4me2 | 32.6 | 28.8 | **51.8** | 34.5 |
> | H3K4me3 | 42.1 | 28.3 | **61.2** | 40.2 |
> | H3K9ac | 57.5 | 49.2 | **63.6** | 52.6 |
> | H3K14ac | 55.0 | 41.6 | **65.6** | 48.0 |
> | H3K36me3 | 63.2 | 47.8 | **65.7** | 53.4 |
> | H3K79me3 |  64.2 | 58.9 | **71.4** | 59.7 |
> | H4 | **82.2** | 77.7 | 79.8 | 79.1 |
> | H4ac | 50.1 | 36.4 | **63.3** | 43.5 |
> | Promoter all | **97.4** | 96.3 | 96.5 | 96.1 |
> | Promoter non-tata | **97.7** | 96.6 | 96.7 | 96.5 |
> | Promoter tata | 96.4 | **96.6** | 96.4 | 96.1 |
> | Splice sites acceptor | **99.0** | 97.6 | 96.6 | 96.6 |
> | Splice sites donor | **98.4** | 98.1 | 96.8 | 96.5 |
> | Splice sites all | **98.3** | 98.0 | 97.9 | 97.3 |
>
> **Table R2. Nucleotide Transformer dataset results. Enhancers and histone mark tasks (names starting with “H”) use the MCC metric. Promoters and splice tasks use the F1 metric, except the “Splice sites all” which uses accuracy.**
>
> ***
>
> ### Species classification (scratch vs. pretrain)
>
> | Length | Scratch |Pretrained |
> |--------|------|------|
> 1k  | 53.9 | 61.1 |
> 32k | 70.7 | 93.4 |
> 250k |  65.7 | 97.9 |
> 450k |  64.5 | 99.4
>
> **Table R3. Species classification ablation, scratch vs. pretraining.**
>
> ***
>
> ### Pretraining GPU & Runtime Comparisons
>
> | Pretrain | DNABERT | Nucleotide Trx | HyenaDNA | HyenaDNA |
> | ----------- | ----------- | ----------- | ----------- |----------- |
> | Params | 110M | 2.5B | 436K | 1.6M |
> | GPUs  | 8-2080 TI  | 128-A100-80GB |  1-A100-40GB  |  1-A100-40GB  |
> | Wall clock | 25 days | 28 days | 80 mins |  80 mins |
> | GPU-hrs | 12,000 | 215,000 | 1.3 | 1.3 |
>
> **Table R4. Pretraining resources and runtime by model. The 436k HyenaDNA model was used on the GenomicBenchmarks while the 1.6M HyenaDNA model was used for the Nucleotide Transformer datasets.**
>
> ***
>
> ### Finetuning GPU & Runtime Comparison
>
> | Model | HyenaDNA | DNABERT | NT-2.5B |
> | ----------- | ----------- | ----------- | ----------- |
> | Parameters | 436 K | 110 M | 2.5 B (0.1% Trainable) |
> | GPUs | 1-A10 | 1-A10 | 1-A10 |
> | Mouse Enhancers | 2s | 32s | 9m 8s |
> | Human Enhancer Cohn | 14s | 9m 25s | 41m 55s |
> | Human Nontata Promoter | 10s | 5m 44s | 35m 36s |
> | Coding vs Intergenomic | 25s | 12m 32s | 1h 40m |
> | Human vs Worm | 25s | 12m 31s | 1h 46m |
> | Human Enhancer Ensembl | 1m 20s | 55m 30s | 2h 56m |
> | Human OCR Ensembl | 1m 32s | 1h 3m | 3h 30m |
> | Human Regulatory | 2m 37s | 1h 46m | 6h 39m |
>
> **Table R5. Finetuning resources and runtime for the GenomicBenchmarks by model.**

---

> > ### Author Response · Authors · 2023-08-10
> >
> > ### Rebuttal 3/3
> >
> > ### Continued:  AttnDNA finetuning on Nucleotide Transformer datasets
> >
> > As suggested by reviewer **[nDFb]**, we finetune the AttnDNA model on the Nucleotide Transformer datasets, as shown in Table R2 above. We sweep a range of (different) hyperparameters for both models. Both AttnDNA and HyenaDNA are causal and use single nucleotide tokens, but AttnDNA significantly underperformed against its Hyena counterpart. This suggests the Hyena operator itself contributes significantly to the overall performance gains of HyenaDNA.
> >
> > This may be confounding, since as reviewer **[nDFb]** noted, Hyena performed similar to attention in language tasks in the original Hyena paper [Poli et. al, 2023]. One reason for the performance gap seen here may be the shallow Transformer size of AttnDNA (2 layers). Shallow Transformers can sometimes underform their convolutional counterparts on small datasets (eg, Vision Transformers vs CNNs on CIFAR-10) [Zhu et al., 2023], which may have more inductive bias spatially or temporally (Hyena does not require positional embeddings for the tokens). The Nucleotide Transformer model is far larger and uses a higher amount of pretraining data, and is able to perform more competitively with HyenaDNA.
> >
> > ### Citations
> >
> > Grešová, Katarína, et al. "Genomic benchmarks: a collection of datasets for genomic sequence classification." BMC Genomic Data 24.1 (2023): 25.
> >
> > Dalla-Torre, Hugo, et al. "The Nucleotide Transformer: Building and Evaluating Robust Foundation Models for Human Genomics." bioRxiv (2023): 2023-01.
> >
> > Poli, Michael, et al. "Hyena hierarchy: Towards larger convolutional language models." arXiv preprint arXiv:2302.10866 (2023).
> >
> > Zhu, Haoran, Boyuan Chen, and Carter Yang. "Understanding Why ViT Trains Badly on Small Datasets: An Intuitive Perspective." arXiv preprint arXiv:2302.03751 (2023).
> >
> > Ji, Yanrong, et al. "DNABERT: pre-trained Bidirectional Encoder Representations from Transformers model for DNA-language in genome." Bioinformatics 37.15 (2021): 2112-2120.

---

> > ### Comment · Reviewer_nDFb · 2023-08-10
> > **Can you share more experiments details to reproduce the results?**
> >
> > Hey,
> >
> > Thank you for preparing the ablation study results! Since the results are too good to be true IMO (much better performance with 1000 times less parameters and much less data, which is extremely rare in the context of deep learning), I tried to reproduce your results on some of the `Nucleotide Transformer` datasets (the ones starting with H), but I am not able to reproduce the similar level of performance as shown here. For example, in the task of `H3K4me3`, the best test set performance I get is MCC: 0.4140 after training for 95 epochs, which is much lower than the 0.612 reported above. I also evaluated on a few randomly selected task (H3, H4, H3K4me2) but fail to reproduce either.
> >
> > It would be really helpful if you can share more experiments details. Or more ideally, it would be excellent if you can provide a python script to reproduce the experiments results.
> >
> > FYI: I download the datasets and randomly split it into train/dev/test (8:1:1). I train the model on 4 V-100-16G GPUs with a per GPU batch size of 64. And the average training time of  a model contains 1 million paramaters per epoch is about 50s. I used the default hyperparamters in the appendix (lr 6e-4, bs 256, epoch 100, weight_decay 0.1, and no rc_aug).
> >
> > Also, as shown in the paper and based on my reproduction, this model often take about 100 epochs to converge, while other works (NT and DNABERT) generally takes 3-5 epochs, which is 20-30 times less. Simply comparing the per epoch training time as above is very unfair. Based on my experiments, in the task of `H3K4me3`, a HyenaDNA model with 1M parameters actually spends slightly more time to fine-tune than a NT with 500M parameters and a DNABERT with 100M parameters. This may differs a bit with different batch sizes and hardwares, but I don't think HyenaDNA is *more efficient* than the baselines in terms of training time and memory usage unless on ultra-long sequences.

---

> > > ### Author Response · Authors · 2023-08-14
> > >
> > > Hi, thank you for your engagement, reviewer **[nDfb]**, and for bringing this up! We’d be happy to help you reproduce the results for the Nucleotide Transformer (NT) datasets.
> > >
> > > We’ll also be sure to provide more details on the exact finetuning procedure and settings in the manuscript for the datasets as well.
> > >
> > > ### To reproduce the HyenaDNA results on the Nucleotide Transformer datasets, you’ll need:
> > >
> > > - the correct model size, **2 layer, d_model=256, sequence_len=1k**, (not d_model=128 as the reviewer used)
> > > - the pretrained weights themselves (for this size), which previously had not yet been shared publicly
> > > - optimized hyperparameters per dataset
> > >
> > > Without having the correct model size (listed in the appendix Table A.3) and a pretrained model itself, there would be a large gap in performance, and likely what you are observing.
> > >
> > > We want to make sure that we account for any differences in code implementations that might be reimplemented online vs what we actually used. This includes accounting for other training settings that are important that may be left out in other demo implementations including: the cosine scheduler, optimizer parameter groups (ie handling hyperparameters per layer config), gradient clip, Pytorch Lightning automatic mixed precision handling, etc.
> > >
> > > As such, we have prepared a **docker image** with the following already on it:
> > >
> > > - an anonymized repo
> > > - the pretrained weights for NT finetuning
> > > - a Dockerfile (or instructions to install dependencies)
> > > - 5 sample NT datasets, preprocessed and split
> > > - a README with the exact commands (and hyperparameters as args) to launch and reproduce (or surpass) the reported results
> > >
> > > **Download repo/data instructions**
> > >
> > > You have 2 options:
> > >
> > > **1. (Easiest w/ Docker):** Pull the docker image with the data, dependencies and pretrained weights ready to go.
> > >
> > > ```
> > > docker pull hyenadna/hyena-dna:latest
> > > ```
> > >
> > > **Run the container**
> > > ```
> > > # gives you an interactive shell (with the dependencies)
> > >
> > > docker run --gpus all -it -p80:3000 hyenadna/hyena-dna /bin/bash
> > > ```
> > >
> > > You’ll end up inside the repo, and there’s a README.md on how to run the experiments (ie the exact launch commands for the 5 sample runs).
> > >
> > > **or,**
> > >
> > > **2. Manual setup:**  Download this anonymous link using `gdown` to obtain the repo. This zip file has the data and pretrained weights inside the repo as well.
> > >
> > > After unzipping the repo, there’s the same README.md (in 1.) about how to reproduce the results. You’ll need to decide how you want to install the dependencies (manually or again with Docker).
> > >
> > > ```
> > > gdown 1ufTP2zMAf1_KkxGxP15PJGNLzfreqpmd
> > > ```
> > >
> > > ### Finetuning time
> > >
> > > There are many ways to look at efficiency, and we appreciate the reviewer providing feedback on this.
> > >
> > > Regarding finetuning time, we’ve provided HyenaDNA convergence results that aggregate the time per dataset. HyenaDNA averaged about 23 mins per dataset, ranging from 1 min to 45 mins, using a single A100 GPU.
> > >
> > > In the Nucleotide Transformer paper itself (section A.3.1), the authors note:
> > >
> > > ```
> > > "On average, a fine-tuning run lasted 20 minutes for the 500M parameter models, and 50 minutes for the 2.5B parameter models.”
> > > ```
> > >
> > > We compared our results to the best performing values from the **NT 2.5B** model, which used 8 A100s (ie, the same GPU type but 8 vs 1 GPU in HyenaDNA).
> > >
> > > Using this baseline, and the NT author reported description, HyenaDNA spends ~17x less GPU-hours finetuning on the same GPU type on the NT datasets.
> > >
> > > For pretraining, where most of the overall compute is spent, the gap between HyenaDNA and the NT 2.5B is even larger, as shown in the Table R4 of the common response.
> > >
> > >
> > > | Dataset | Pretrain (min) | Scratch (min) |
> > > |------|------|------|
> > > | Enhancer | 8 | 6 |
> > > | Enhancer types | 8 | 10 |
> > > | H3 | 10 | 5 |
> > > | H3K4me1 | 45 | 16 |
> > > | H3K4me2 | 40 | 50 |
> > > | H3K4me3 | 40 | 60 |
> > > | H3K9ac | 30 | 13 |
> > > | H3K14ac | 45 | 50 |
> > > | H3K36me3 | 20 | 45 |
> > > | H3K79me3 |  40 | 15 |
> > > | H4 | 1  |  5 |
> > > | H4ac | 30 | 55 |
> > > | Promoter all | 40 | 20 |
> > > | Promoter non-tata | 3 |10 |
> > > | Promoter tata | 2 | 2 |
> > > | Splice sites acceptor | 50 | 40 |
> > > | Splice sites donor | 20 | 35 |
> > > | Splice sites all | 30 | 30 |
> > > | Average | 23 | 26 |
> > >
> > > **Table R6. Time to convergence for HyenaDNA finetuning and training from scratch on the Nucleotide Transformer datasets.**
> > >
> > > Reviewer **[DWDg]** also suggested comparing the (pretrained) finetuning vs scratch training times on the NT datasets for HyenaDNA. Though they have similar training times for reaching their respective top metrics (23 vs 26 mins on average per dataset), the pretrained models generally reach the same performance metric (as the scratch models) faster, and then continue to eke out more gains, albeit more slowly. We appreciate the suggestion and will put these results in the appendix of the manuscript.

---

> > > > ### Comment · Reviewer_nDFb · 2023-08-17
> > > > **Able to reproduce the results with the provided code and model.**
> > > >
> > > > Thank you very much for preparing the docker image. Here I can confirm that with the provided codes and pre-trained weight, I am able to reproduce similar-level of results on H3K4me2 (44.7) and H3K14ac (64.8). Now I have no doubt with the evaluation results.
> > > >
> > > > Though the comparison with NT is a bit unfair (There is no validation set, so the hyper-parameters tuning and best model selection are directly based on test set), the result is still very impressive given the tiny size of the model.
> > > >
> > > > However, I am still confused about the great performance of the pre-trained model. Can you provide more details in the difference between the model you shared in the docker image v.s. those publicly available checkpoint? I am asking since my own experiments show that using those publicly available models and my own reproduction to replace the checkpoint you provided leads to very dramatically performance drop (e.g., 60 -> 40). If the main beneficial of the your comes from the Hyena operator, I do not understand why the this one is much more powerful than others given the fact that they have similar architecture and number of parameters.
> > > >
> > > > Is it because the pre-training is unstable and you are not able to consistently get a good model with different size or you apply different strategy and tricks to this specific model? Is is possible that there's some data leakage issue on this checkpoint? (I do not suspect you at all but just provide a possibility here.)
> > > >
> > > > I think it would be really beneficial to the community if you can discuss these in more details in your paper. If you can provide the pre-training code for us to reproduce the pre-training stage, it would be the best. I understand this needs a lot of work, but I think we are responsible to make sure everything is correct before recommending a paper with such exceptional performance.

---

> > > > > ### Author Response · Authors · 2023-08-17
> > > > >
> > > > > Thank you! We very much appreciate the level of engagement from the reviewer, it’s helping us improve the manuscript and communicate the work better.
> > > > >
> > > > > ### NT comparison
> > > > >
> > > > > Indeed, the NT uses a validation set, however, there are 2 sets of results in their paper. One where they use the validation set hyperparameters, and one where they do not.
> > > > >
> > > > > In the first set of results in **Figure 2**, the validation set is used to select the hyperparameters for the test set. However, in the NT **Supplementary Table 6**, these results reflect the **best test results overall across all hyperparameters - regardless of validation set hyperparameters**.
> > > > >
> > > > > We compare with the best NT scores overall, from Supplementary Table 6, which are the highest test scores regardless of validation hyperparameters, and a more fair comparison with our results.
> > > > >
> > > > > ### Pretrained model difference
> > > > >
> > > > > From what the reviewer described about using an outside checkpoint (outside the Docker image we provided), the main reason is the ***different model size***, ie, they do not have the same number of parameters.
> > > > >
> > > > > Specifically, the reviewer appeared to use a model with **2 layers, and d_model=128, which has 436K parameters**.
> > > > >
> > > > > The model (in the Docker) that we actually finetuned on the NT datasets is **4x** the size, using **2 layers, d_model=256, and has 1.6M parameters**. This model size is reported in section A.2.2. in the appendix. We see that there may be confusion on model size for the NT datasets, and so we will be sure to highlight this more clearly in the manuscript.
> > > > >
> > > > > Note: the model size scales quadratically in model dimension / width (ie, 2x the d_model, means 4x the number of parameters). Also, as mentioned in the previous common response on Aug. 13, this model was previously not released publicly, and likely different than what the reviewer used.
> > > > >
> > > > > ### Pretraining code
> > > > >
> > > > > Thank you for the suggestion. We agree with the reviewer that providing more details about the pretraining and releasing the code publicly will help the community. Fortunately, the Docker image we shared does already have the exact pretraining code we used. This code will be part of our public release plan, as well as the hyperparameters.
> > > > >
> > > > > We’ve also provided instructions for the reviewer if they’d like to reproduce the pretrained weights (for the NT finetuning) and see for themselves. **(Using the previous Docker image)**
> > > > >
> > > > > **1. Download** the data via `gdown` into the `/wdr/data/hg38` directory of the Docker image repo.  (For convenience, we’ve uploaded this anonymously for the reviewer)
> > > > >
> > > > > ```
> > > > > # from the repo root, download and unzip
> > > > > mkdir /wdr/data/hg38 && cd /wdr/data/hg38
> > > > > gdown 1MDYoSU7mb8ReS7trL36x03b5RDo9KJhc && gdown 1U3U8UZWLy37ycb0DublSbcsyZby1Rvt4
> > > > > gunzip hg38.ml.fa.gz && cd /wdr
> > > > > ```
> > > > >
> > > > >
> > > > > **2. Launch** (from `/wdr`)
> > > > >
> > > > > ```
> > > > > python train.py wandb=null experiment=hg38/hg38_hyena model.n_layer=2 model.d_model=256 dataset.bed_file=/wdr/data/hg38/human-sequences.bed dataset.fasta_file=/wdr/data/hg38/hg38.ml.fa model.fused_dropout_add_ln=True dataset.batch_size=256 dataset.max_length=1024 dataset.max_length_val=1024 trainer.precision=bf16
> > > > > ```
> > > > > This should reproduce the same pretrained weights we used for the NT. It reaches a test loss ~1.16 in about 80 mins on a single A100.
> > > > >
> > > > >
> > > > > ## Explicitly, our plans after the review period, we will:
> > > > >
> > > > > - publicly release pretrained weights (including the exact model for the NT finetuning)
> > > > > - release all code for pretraining and finetuning (our full implementation)
> > > > > - discuss further details for pretraining and finetuning in order to reproduce results in the appendix
> > > > > - provide the Dockerfile and image to make it even easier for the community to test our code
> > > > >
> > > > > ### Differences in code
> > > > >
> > > > > In accordance with the double blind review process - we can only speculate how our code (in the Docker image) may differ from other implementations of HyenaDNA online, or your own reimplementation. To reiterate our comment on this in our previous common response on Aug 13th, differences with other **simplified demonstration implementations online** may include:
> > > > >
> > > > > - differences in model size settings
> > > > > - lack of cosine decay learning rate scheduler
> > > > > - lack of optimizer parameter groups, that allow allow for different hyperparameters per layer (eg, the Hyena layer requires weight decay to be 0)
> > > > > - using gradient clip (with value of 1.0)
> > > > > - automatic mixed precision handling in Pytorch Lighting (eg, not available on colab)
> > > > >
> > > > > Demonstration code online is best suited for understanding the core architecture components. For reaching SotA, more sophisticated training code is required to reach a model's full potential. For HyenaDNA, we'll be sure to share these tips in greater detail of the manuscript.
> > > > >
> > > > > We thank the reviewer for their valuable feedback, and as always, we’re happy to provide any further clarifications the reviewer might have.

---

> > > > > > ### Comment · Reviewer_nDFb · 2023-08-19
> > > > > > **Able to reproduce the results by pre-trained a model from scratch.**
> > > > > >
> > > > > > Thanks for providing the instruction for model pre-training! Here I can confirm that will the provided code, I can pre-train a model from scratch and achieve similar-level performance on the selected downstream tasks.
> > > > > >
> > > > > > Though we still do not fully understand where how this model is able to be so effective given its size, I think the empirical results suggest that HyenaDNA is a very solid milestone work in the area of genome foundation model. I have raised my score from 4 to 7.

---

> > > > > > > ### Author Response · Authors · 2023-08-19
> > > > > > >
> > > > > > > We're glad the reviewer was able to reproduce the pretrained model using thing the provided code, and reach similar downstream performance.
> > > > > > >
> > > > > > > We very much thank the reviewer for their lively discussion and sharp questioning, and appreciate the upgraded score. We're super optimistic about this line of work, and will certainly continue driving its applications. This includes more empirical support as well as diving deeper into understanding the mechanisms that enable its effectiveness.
> > > > > > >
> > > > > > > Let us know if there are any other concerns you might have, and thanks again!

---

### Decision · Program_Chairs · 2023-09-21

**Decision:**

Accept (spotlight)

**Comment:**

The reviewers agree that the authors' methods achieve strong empirical performance over previous methods, the approach appears novel, and the paper is well written. In response to the reviewers' common ask for a variety of ablation studies, the authors responded with the results of these studies and all reviewers were satisfied with the response. Ultimately, this paper represents a fairly significant engineering success: a foundation model trained on DNA that achieves state of the art performance with a much smaller model but much larger context window. Practitioners working with genomic sequence data are sure to find such a model very valuable.